# Biomimetic KcsA channels enabled by 1D MOF-in-2D COF

Qian Sun, Pengjia Dou, Jingcheng Du, Ayan Yao, Dong Cao, Ji Ma, Shabi Ul Hassan, Jian Guan & Jiangtao Liu ✉

Highly permeable and selective biomimetic membranes that can feel and recognize valuable ion species have attracted enthusiastic interest due to their analogous behavior with biological ion channels and potential applications in rigorous ion sieving. However, designing and developing single-species selective membranes that can isolate monovalent cations such as $K^+$/$Na^+$ remains a tremendous challenge due to the sub-nanometer ion size, as well as the angstrom-sized difference. Considering the non-homogeneous hetero-structure of KcsA channels and -COOH groups generally showing lower $K^+$ affinity, we propose the 1D MOF (rich in -COOH groups)-in-2D COF concept, aiming to enhance $K^+$/$Na^+$ separation through strategic construction of heterogeneous ion transport channels, therefore narrowing the pore size of pristine COF membrane, and weakening the $K^+$-channel wall interactions. Concretely, by interlocking and in situ immobilized growth, the pristine COF membrane is capable of capturing MOF ligands and metal ions in sequence to form 1D MOF-in-2D COF hetero-structured composite membranes. Benefiting from the molecular-level interlinked hybridization of covalent and metal organic hetero-frameworks induced by the coordination interaction between the -NH groups in COFs and the Cu centers from MOFs, the composite membrane enables rapid diffusion of $K^+$ in confined heterogeneous channels, thus leading to unprecedented cation sieving performance with $K^+$/$Na^+$ selectivity approaching $10^2$ and $K^+$/$Mg^{2+}$ selectivity exceeding $10^3$. This membrane design concept exploits a viable avenue for developing single-species selective biomimetic membranes to achieve ultrahigh separation performance.

Two-dimensional (2D) membranes with sub-nanometer or sub-2-nanometer confined channels have attracted extensive interest during the last decade due to anomalous and unexpected ion transport behaviors[1–4]. In a confined space, the influence of channel wall-ion interactions, which can be ignored at a large scale, is significantly enhanced, and becomes an important or even dominant factor affecting the ion transport process[5,6]. Proverbially, KcsA channels with the size of ~5.6 Å are capable of strictly regulating ion transport and exchange across a narrow, heterogeneous, 12-Å-long selectivity filter[7–9]. Based on the nano-confinement effect and channel wall-ion interactions (carbonyl oxygen atoms lining the walls of the pore coordinate $K^+$ but not smaller $Na^+$ after removing their solvation shell), the biological channels reliably exclude the smaller alkali metal cations $Li^+$ (radius 0.60 Å) and $Na^+$ (0.95 Å) but allow larger members of the series $Rb^+$ (1.48 Å), $Cs^+$ (1.69 Å), and $K^+$ (1.33 Å) to pass through to near-diffusion limited rates, thus, resulting in an ultrahigh $K^+$/$Na^+$ selectivity of $10^4$[10–13].

Inspired by natural ion channels, a number of delicately engineered biomimetic membranes have been designed over the past few

State Key Laboratory of Advanced Environmental Technology, Department of Environmental Science and Engineering, University of Science and Technology of China, Hefei 230026, China. ✉e-mail: jiangtaoliu@ustc.edu.cn

decades[1–4]. A multilayered polymer membrane embellished with iminodiacetate (IDA) functional groups was skillfully developed by DuChanois et al.[2]. Channel wall-ion interactions are responsible for the high-precision ion separation, and metal ions with higher binding energy to IDA groups pass through the membrane more readily, whereas weaker binding species have a markedly decreased permeability. Consequently, the fabricated polymer membrane shows high selectivity for the cations with the same valence state and almost identical hydration diameter such as $Cu^{2+}/Co^{2+}$ (>30), but the separation performance for $K^+/Na^+$ (also with the same valence state and almost identical hydration diameter) is not investigated[2]. A structured metal-organic framework-based sub-nanochannel (MOFSNC) has also been prepared and enables an ultrahigh mono/divalent ion selectivity of $10^3$, which is attributed to the ion-carboxyl interactions substantially lowering the energy barrier required for monovalent cations to pass through[3]. A COF (covalent organic framework)-based membrane with orderly aligned oligoethers in the pore channels affords a high $Li^+/Mg^{2+}$ separation factor of 64 due to the accelerated transport of $Li^+$ by weakened channel wall-$Li^+$ ion interactions[4]. Despite advances in the development of highly permeable and selective biomimetic membranes capable of recognizing and separating valuable ion species, research efforts have largely focused on the regulation of channel wall-ion interactions, and designing and developing biomimetic KcsA channels remains a formidable challenge. Note that the four protein subunits that comprise the biological potassium channels are four-folded symmetric heterotetrameric complexes, in addition to the confined transport, short selectivity filter, and channel wall-ion interactions[10]. This implies that the heterostructure and non-homogeneous channels may be one of the indispensable factors to achieve ultrahigh $K^+/Na^+$ selectivity.

As a type of crystalline porous membranes, 2D COF membranes originating from the atomically accurate integration of building components through covalent bonds have garnered significant attention in recent decades[14,15]. The well-organized one-dimensional (1D) nanochannels, readily customizable functions, and tunable aperture arrangement make them excellent candidates for developing hetero-structured composite membranes. A cysteine functionalized COF membrane (COF-Cys) was developed by effectively anchoring amino acids on the channel walls to serve as the ion-selective switches, allowing for switchable $Na^+/K^+$ selectivity in a single membrane by altering solution pH, resulting in a $K^+/Na^+$ selectivity of 1.7 at pH 3.8 and $Na^+/K^+$ selectivity of 2.9 at pH 8.9[16]. Fan et al. proposed a classic MOF-in-COF concept for the confined growth of three-dimensional (3D) unit cell-sized MOFs inside a supported 2D COF layer to prepare MOF-in-COF molecular sieving membranes[17]. With the combined advantages of fast molecular transport channels and accurate size sieving, the membrane demonstrated superior performance in terms of ultrahigh $H_2$ permeance and separation selectivity for $H_2/CO_2$ and $H_2/CH_4$. It is worth noting that, in contrast to the 3D MOFs in the 1D channel of 2D COFs where the continuous MOF skeleton is destroyed, anchoring chain-like 1D MOFs in the 1D channel will be more advantageous and logical to fabricate orientated MOFs inside the confined COF channels.

In this contribution, we report a series of 1D MOF-in-2D COF hetero-structured composite membranes prepared by a two-phase interfacial polymerization (IP). The chain-like 1D MOFs can be orientally (in the (1 0 −1) plane) developed inside the 1D channels of the 2D COF membrane benefiting from the nano-confined effect, ligand trapping, and interlocking growth process, therefore achieving the strategic construction of heterogeneous ion transport channels with narrowed pore size and weakened $K^+$-channel wall interactions (Fig. 1). The prepared composite membrane presents a molecular-level interlinked hybridization of covalent- and metal- organic frameworks, which is induced by the coordination between the -NH groups in COFs and the Cu centers from MOFs. Most importantly, the 1D MOF-in-2D COF composite membrane exhibits unprecedented single-species selective properties similar to KcsA channels with ultrahigh $K^+/Na^+$ selectivity of 82.52 and $K^+/Mg^{2+}$ selectivity of 1131.07.

## Results

### Oriented CMOF composite membrane

As a proof of concept, 2D TAPA-TFP COF membranes with adjustable thickness (Supplementary Fig. 1a−d, Supplementary Data 1), rigid skeleton, superior tenacity and flexibility, and vertical 1D channels are chosen as the solid scaffolding and nano-confined template for composite membrane fabrication (Fig. 1)[18]. To achieve the molecular-level interlinked hybridization of covalent- and metal- organic frameworks,

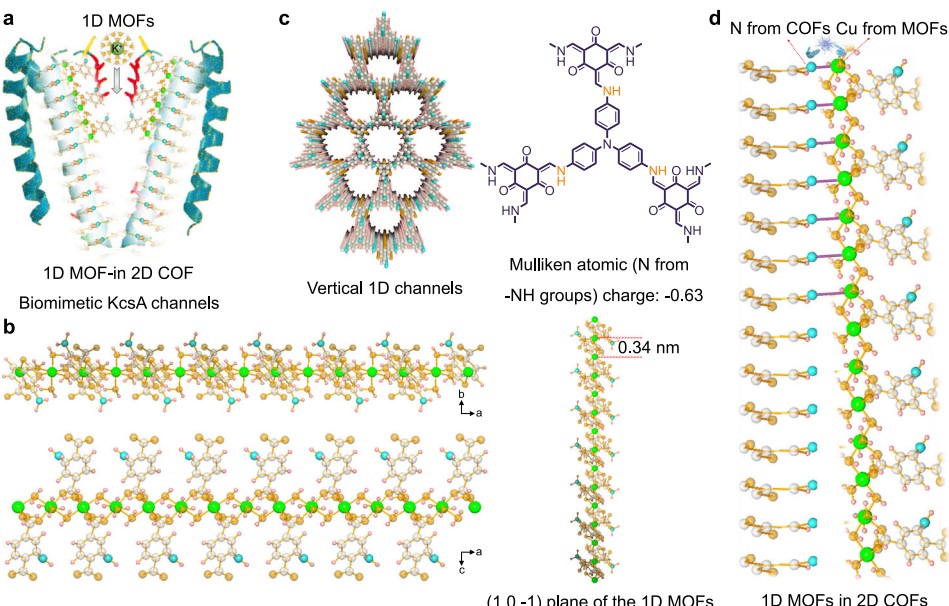

**Fig. 1 | 1D MOF-in-2D COF hetero-structure. a** Schematic illustration of the 1D MOF-in 2D COF biomimetic KcsA channels. **b** Structure of the 1D MOFs. **c** Vertical 1D channels and Mulliken atomic (N) charge of TAPA-TFP COFs. **d** Hetero-structure of the 1D MOFs in 2D COFs. Apricot, sapphire, pink, cyan and silvery spheres represent O, N, H, Cu and C atoms, respectively. Part of the COF structure is omitted for clarity.

the chain-like 1D $NH_2$-CuBDC MOFs featuring accessible $Cu^{II}$ sites and coordination-exchange characteristics are selected as a showcase (Fig. 1b, Supplementary Fig. 1e, Supplementary Data 2)[19]. Mulliken population analysis (Fig. 1c, Supplementary Table 1) indicates that electron-rich regions are localized over N atoms of TAPA-TFP COFs and the N atoms from -NH groups with a Mulliken atom charge of −0.63 are inclined to coordinate with the Cu centers from MOFs due to the lower steric hindrance (Fig. 1d). Simulation results indicates the short-range coordination interaction between MOFs and COFs (Supplementary Fig. 1f).

Typically, after the successful preparation of the self-standing TAPA-TFP COF membrane at the interface, $NH_2$-BDC is slowly injected into the underlying oil phase by an injection syringe to allow the -$NH_2$ of $NH_2$-BDC to fully react with the -CHO at the COF defects (Note that research and simulation have confirmed that regardless of how perfect the crystal growth and stack, topological defects of COFs are inevitable) (Fig. 2a)[20]. Furthermore, $NH_2$-BDC can be captured and in-situ immobilized into the 1D COF channels waiting for the coordination with $Cu^{II}$ atoms due to van der Waals and hydrogen bonding interactions (Fig. 2b). Subsequently, $Cu(NO_3)_2$ is gently injected into the aqueous phase. Due to its small hydrated diameter (0.8 nm) and capillary effect, $Cu^{2+}$ can easily drill into the channels and coordinate with $NH_2$-BDC. The obtained composite membrane is denominated as TAPA-TFP-$x$-$NH_2$-CuBDC CMOF membrane, with $x$ being the concentration of $NH_2$-BDC (Supplementary Fig. 2). The TAPA-TFP-$x$-$NH_2$-CuBDC CMOF membrane with superior mechanical properties shows a Janus morphology and its bottom surface is uniformly distributed with broken thin COF vesicles that nearly occupy the surface, in contrast to the pristine COF membrane (almost identical top and bottom surface morphologies) (Fig. 2c, Supplementary Fig. 2, Supplementary Movie 1). With the hierarchical increase of MOF ligand concentration, the thickness of the CMOF membrane progressively rises (Supplementary Fig. 2). The multifold increase in thickness suggests that the MOFs can grow into the interlayer or surface of the pristine COF membrane, or the insertion of MOFs can promote the secondary growth of the pristine COF membrane. Surprisingly, we failed to find definite MOFs on the top and bottom surfaces of CMOF membranes (Supplementary Fig. 3). The uniform Cu signals from energy-dispersive X-ray spectroscopy (EDXS) suggest a good dispersion of MOFs in the CMOF membranes (Fig. 2c, Supplementary Fig. 4)[17]. Noteworthy, individual copper ions from $Cu(NO_3)_2$ cannot coordinate with COFs (Supplementary Fig. 5). HRTEM (high-resolution transmission electron microscope) images reveal the interlinked covalent- and metal- organic hetero-frameworks, and the lattice fringes from both MOFs (0.27 nm for the (5 0 -5) plane) and COFs (0.39 nm for the (0 0 1) plane) are observed (Fig. 2d, Supplementary Fig. 6). The TAPA-TFP-0.25-$NH_2$-CuBDC CMOF composite membrane features a main pore diameter of 0.68 nm (basically consistent with simulation results), which is smaller than that of COFs (1.22 nm) or MOFs (1.42 nm), also demonstrating the molecular-level interlinked hybridization frameworks (Fig. 2e, Supplementary Fig. 7a–e). It is noteworthy that the CMOF composite membrane exhibits another weaker peak at 1.20 nm, which is similar to the pore size of pristine COF membrane. This suggests that the chain-like MOFs may not fully occupy the COF pore channels in a complete top-down filling manner (Supplementary Fig. 7f). Computation results indicate that in the complex confined channels with a diameter of 0.68 nm, although the -COOH groups carry significant charges in an aqueous solution environment, they are almost unable to combine water molecules to form stable hydrated layers. Therefore, the pore size obtained from BET can represent the effective aperture for ion transport (Supplementary Fig. 7g). Moreover, the CMOF composite membrane displays a smaller surface area (88.24 $m^2 g^{-1}$) in comparison with the pristine COF membrane (307.41 $m^2 g^{-1}$) due to the presence of MOFs in the 1D channels. The actual MOF weight loadings in CMOF

membranes are confirmed by a thermogravimetric analyzer (Supplementary Figs. 8, 9).

XPS (X-ray photoelectron spectroscopy), UV-Vis-NIR, XANES (synchrotron-based X-ray absorption near edge structure) and ATR-FTIR (attenuated total reflection-Fourier transform infrared microscope) confirm that different from the reported COF-MOF hybrids where definite COFs or MOFs can be observed and core-shell or coating structures can be achieved by a simple combination of COFs and MOFs[21], the prepared TAPA-TFP-$x$-$NH_2$-CuBDC CMOF composite membrane presents a molecular-level interlinked hybridization of covalent- and metal- organic frameworks, which is induced by the coordination between the -NH groups in COFs and the Cu centers from MOFs (Supplementary Figs. 10–14, Supplementary Table 2)[17,22]. The slight difference in surface wettability between the pristine COF membrane and CMOF composite membranes results from the introduction of MOFs (Supplementary Fig. 15). Close examinations of the XRD peak position and intensity and GIWAXS (grazing incidence wide angle X-ray scattering) confirm that the TAPA-TFP-5-$NH_2$-CuBDC CMOF membrane is preferentially oriented and the diffraction peaks at 6.1° and 12.2° are respectively assigned to the peaks from the (1 0 −1) and (2 0 −2) crystallographic planes of MOFs (Figs. 1b and 2f). When reducing the concentration of $NH_2$-BDC or increasing the thickness of the pristine COF membrane, the XRD peaks of MOFs almost disappear, suggesting that the XRD peaks of MOFs are easily obscured after growing into COFs (Supplementary Fig. 16). It is worth mentioning that the sharp XRD peaks of the pristine COF membrane that should have been detected (The ordered and definite structure of the COF membrane had already been established before forming the CMOF membrane as displayed in Fig. 2a.) are not distinctly observed in the CMOF composite membrane, which may result from: 1) the introduction of MOFs into the pore structure or interlayer of COFs based on coordination interactions masking the XRD signals from COFs, and 2) the intense XRD signals from MOFs overwhelmingly suppressing the COF signals, making the XRD peaks from COFs extremely difficult to observe when MOFs and COFs coexist[17,23]. Still, the (110) crystallographic plane with weak signals of pristine COF membrane can be observed in CMOF composite membrane (Fig. 2f), and its right-shift compared with pristine COF membrane confirms the effective growth of MOFs within the COF channels and the relatively strong interaction between MOFs and COFs. When increasing the thickness of the pristine COF membrane or changing the distribution of MOF monomers in two phases during the IP process, the obtained CMOF membranes exhibit comparable characteristics and definite MOFs are still not observed (Supplementary Figs. 17–23, Supplementary Table 3). After the introduction of MOFs, the COF vesicles/nanotubes distributed in the bottom surfaces of membranes change from plump to shriveled, which results from the interfacial COF membrane hindering heat (generated from the MOF growth process) dissipation across the water/oil interface and the heat accumulation leading to local fracture and shriveling of pristine COF vesicles/nanotubes[24].

## MCOF composite membrane

Generally, after the successful preparation of $x$-$NH_2$-CuBDC MOFs at the oil-water interface, TFP dissolved in dichloromethane was slowly injected into the underlying oil phase by an injection syringe to allow the -$NH_2$ of MOFs to fully react with the -CHO of TFP (this movement should be as slow and gentle as possible to avoid interface disturbance). One day later, TAPA dissolved in dichloromethane was slowly injected into the underlying oil phase to initiate the Schiff base reaction. After three days of reaction, $x$-$NH_2$-CuBDC-TAPA-TFP MCOF membranes can be observed at the oil-water interface, with $x$ being the concentration of $NH_2$-BDC (Fig. 3a). On the noticeably smoother top surfaces, unpenetrated crater-like structures appear, which may result from the MOF lamellae retarding heat dissipation, leading to the local

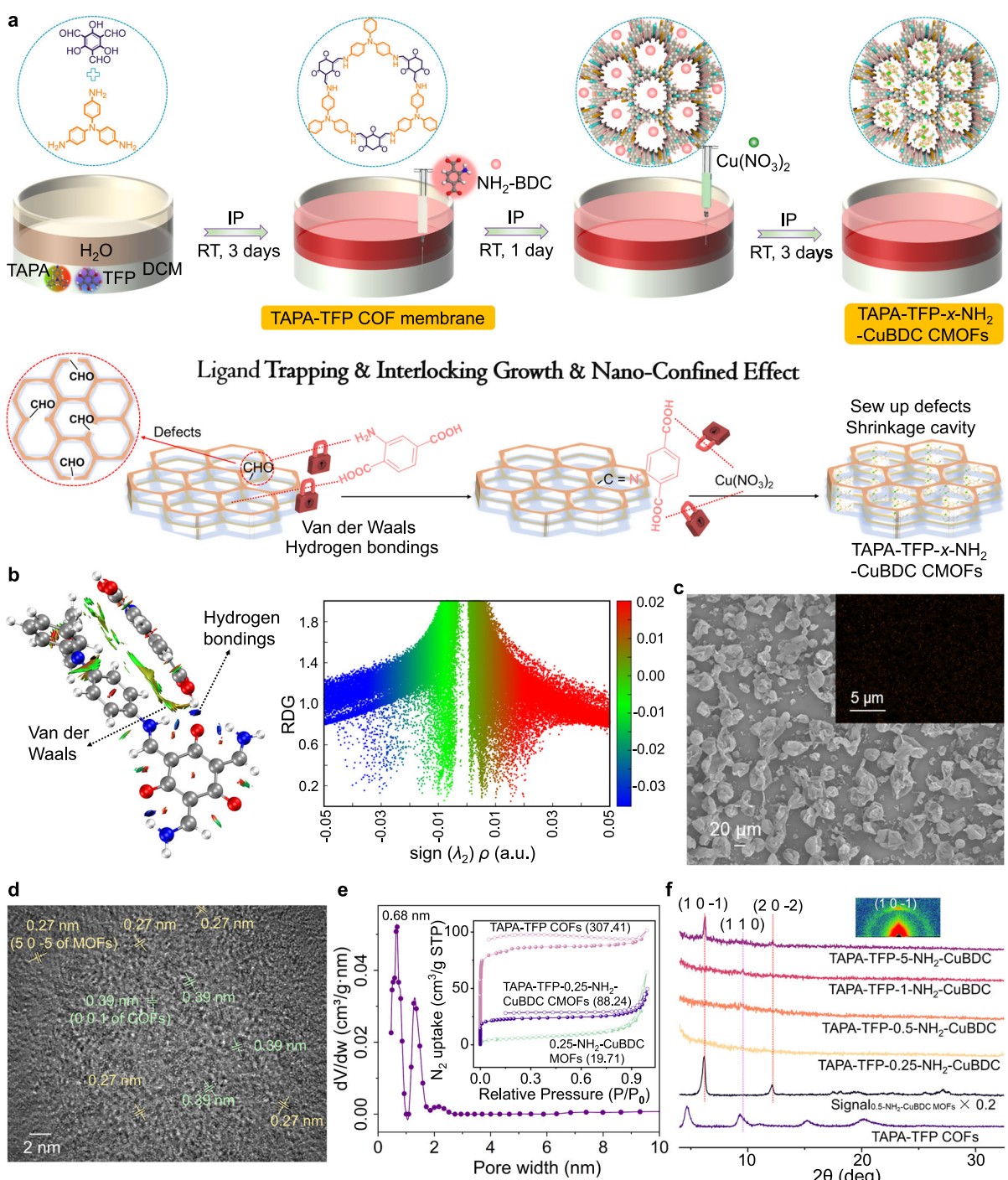

**Fig. 2 | Oriented CMOF composite membrane fabricated by interlocking and in-situ immobilized growth. a** Schematic illustration of the growth process of CMOF composite membranes. **b** Quantum theory of atoms in molecules (QTAIM) topology analysis with reduced density gradient (RDG) isosurface for the visualization of noncovalent interaction (NCI) for TAPA-TFP COFs and NH$_2$-BDC complexes in a spatial region. Green and blue isosurfaces indicate van der Waals and hydrogen bonding interactions, respectively. **c** SEM image of the bottom surface of the TAPA-TFP-0.25-NH$_2$-CuBDC CMOF composite membrane (inset: Corresponding EDXS mapping and Cu elemental distribution, COFs: 0.5 mM TAPA + 0.5 mM TFP). HRTEM image (**d** COFs: 0.5 mM TAPA + 0.5 mM TFP) and pore size distribution (**e** inset: Brunauer-Emmett-Teller (BET) surface areas, filled symbols for adsorption and unfilled symbols for desorption, COFs: 1 mM TAPA + 1 mM TFP) of the TAPA-TFP-0.25-NH$_2$-CuBDC CMOF composite membrane. **f** XRD profiles of prepared CMOF composite membranes (inset: GIWAXS pattern of the TAPA-TFP-5-NH$_2$-CuBDC CMOF composite membrane, COFs: 0.5 mM TAPA + 0.5 mM TFP).

formation of gas nanobubbles (Fig. 3b, Supplementary Figs. 24, 25)[24,25]. Different from the CMOF membranes, with the hierarchical increase of MOF ligand concentration, the thickness of the MCOF membrane progressively decreases, which is attributed to the inhibition effect of MOFs at the interface on the out-of-plane stacking of COF membrane (Supplementary Fig. 24). Expectedly, MOF lamellae are observed on both the top and bottom surfaces of MCOF membranes and the Cu signals from EDXS confirm a good distribution of MOFs in the MCOF composite membranes, which suggests that the TAPA-TFP COFs can capture and suck the MOF lamellae at the interface to grow integrally (Fig. 3b, Supplementary Figs. 26, 27). Regarding the pore size distribution, the pristine COF membrane exhibits sharp peaks centered at

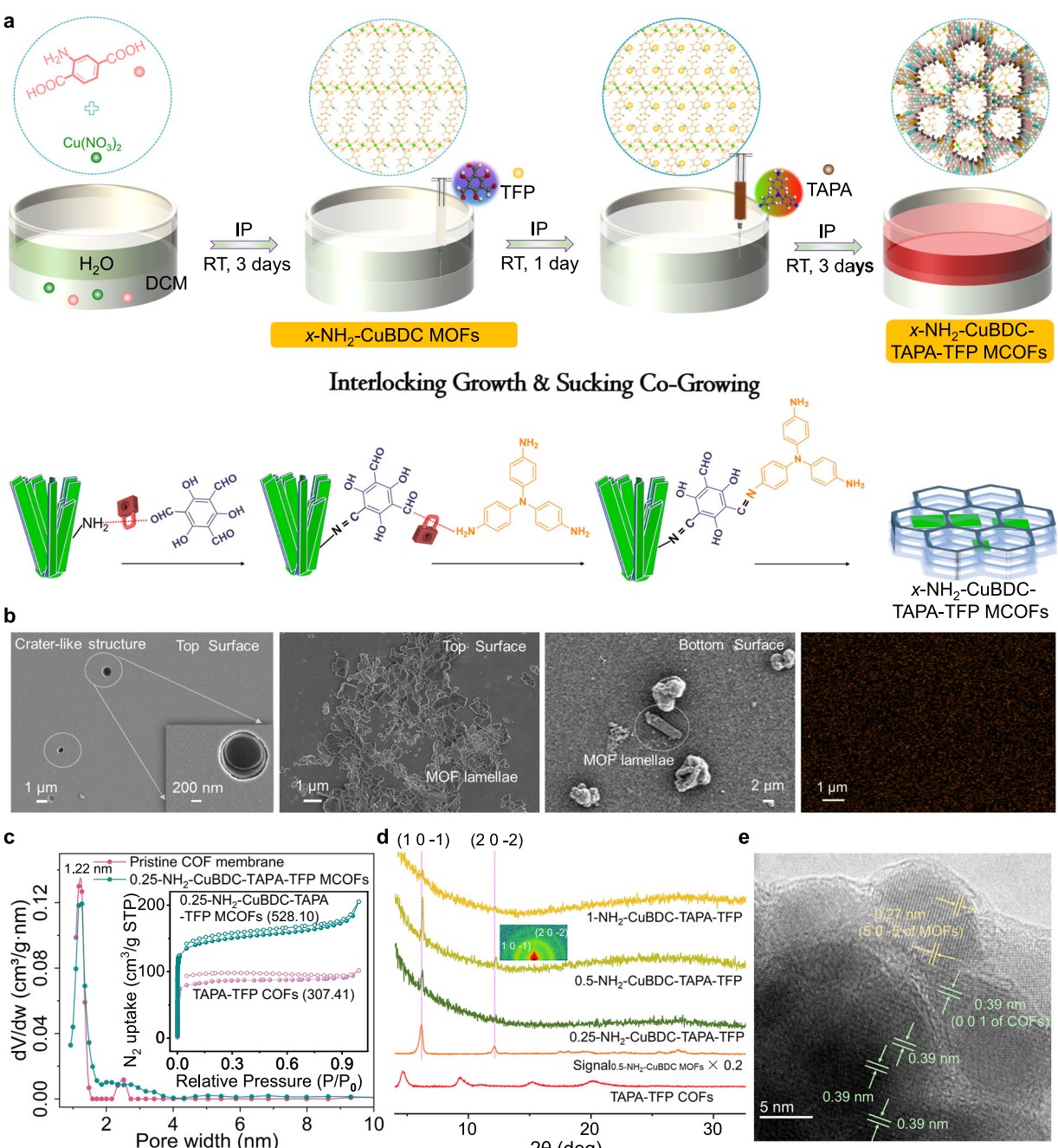

**Fig. 3 | MCOF composite membrane fabricated by trapping and sucking co-growing. a** Schematic illustration of the growth process of MCOF composite membranes. **b** SEM images of the 0.5-NH₂-CuBDC-TAPA-TFP MCOF composite membrane and corresponding EDXS mapping and Cu elemental distribution. **c** Pore size distribution (inset: BET surface areas, filled symbols for adsorption and unfilled symbols for desorption) of the 0.25-NH₂-CuBDC-TAPA-TFP MCOF composite membrane. **d** XRD profiles of prepared MCOF composite membranes (inset: GIWAXS pattern of the 0.25-NH₂-CuBDC-TAPA-TFP MCOF composite membrane). **e** HRTEM image of the 0.5-NH₂-CuBDC-TAPA-TFP MCOF composite membrane.

1.22 and 2.52 nm, while 0.25-NH₂-CuBDC-TAPA-TFP MCOF composite membrane shows only one peak at 1.22 nm (Fig. 3c). The disappearance of larger pores (2.52 nm) from inevitable topological defects is attributed to the insertion and suturing function of MOFs. Non-significant pore size changes compared to pristine COF membranes imply that the molecular-level linkage between MOFs and COFs in the MCOF membrane is very weak and the post-MOF synthesis strategy is more favorable to the formation of cross-linked heterogeneous frameworks (Figs. 2e and 3c). The MCOF composite membrane exhibits a significantly larger surface area (528.10 m² g⁻¹) than the pristine COF membrane (307.41 m² g⁻¹), due to the MOFs at the interface inhibiting heat loss and the higher temperature increasing the porosity of COF membrane. Interestingly, 0.5-NH₂-CuBDC-TAPA-TFP MCOF composite membrane exhibits the sharpest XRD peaks (Fig. 3d), and the HRTEM detected the lattice fringes from both MOFs (0.27 nm for the (5 0 −5) plane) and COFs (0.39 nm for the (0 0 1) plane) (Fig. 3e). The actual weight loading of MOFs in the MCOFs is generally higher than that of CMOFs (Supplementary Fig. 28). ATR-FTIR, XPS, UV-Vis-NIR and XANES demonstrate the successful insertion of MOFs but the very weak and even negligible coordination interactions between

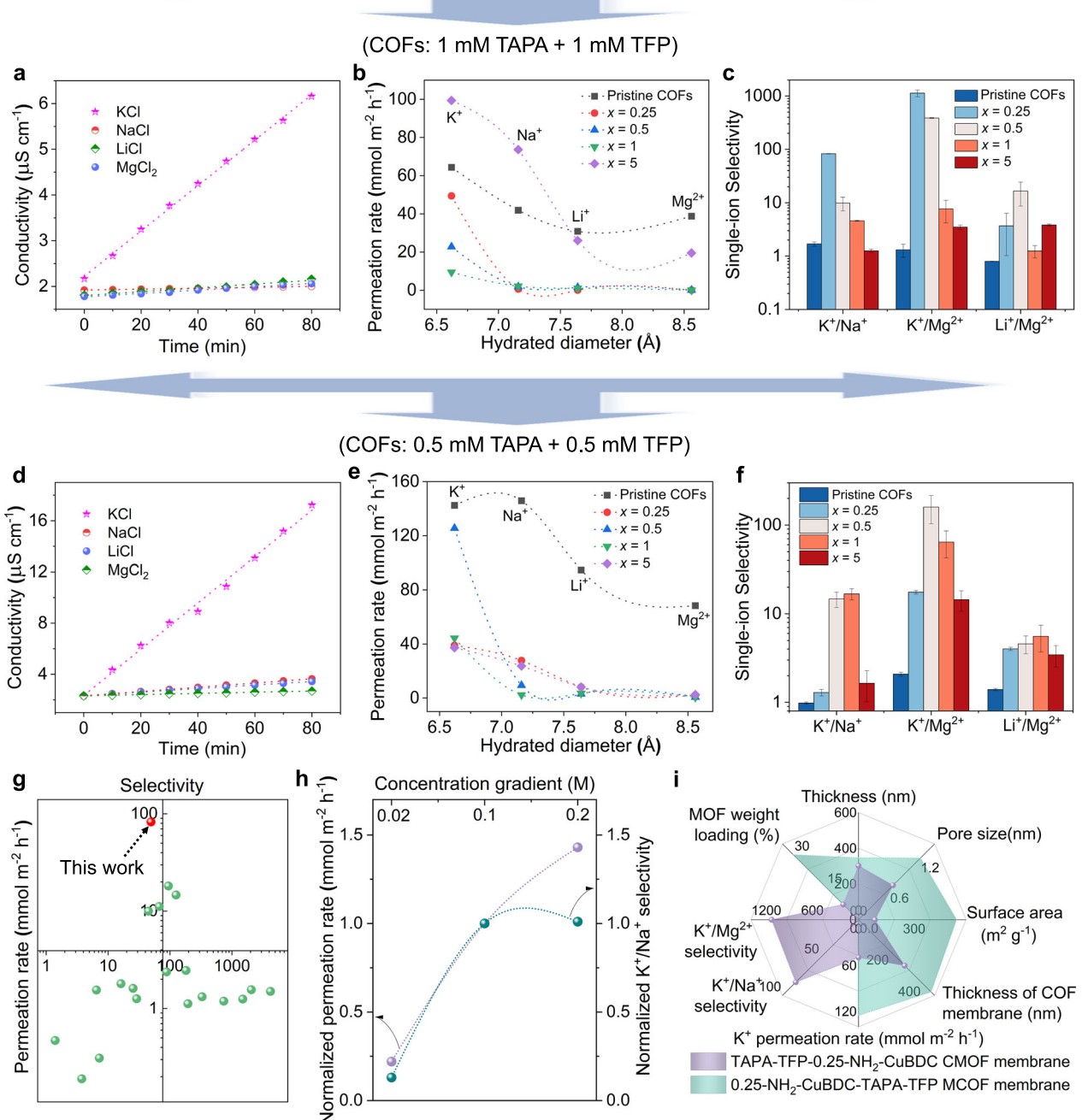

**Fig. 4 | Ion permselectivity of CMOF composite membranes. a** Cation diffusion behavior of the TAPA-TFP-0.25-NH$_2$-CuBDC CMOF composite membrane (COFs: 1 mM TAPA + 1 mM TFP, linear fitting is employed, and the R$^2$ values for KCl, NaCl, LiCl, and MgCl$_2$ plots are 0.999, 0.995, 0.979 and 0.996 respectively). Ion permeation rate (**b**) and single-ion selectivity (**c**) of TAPA-TFP-$x$-NH$_2$-CuBDC CMOF composite membranes (COFs: 1 mM TAPA + 1 mM TFP, $x$ = 0.25, 0.5, 1, 5). **d** Cation diffusion behavior of the TAPA-TFP-0.5-NH$_2$-CuBDC CMOF composite membrane (COFs: 0.5 mM TAPA + 0.5 mM TFP, linear fitting is employed, and the R$^2$ values for KCl, NaCl, LiCl, and MgCl$_2$ plots are 0.994, 0.999, 0.999 and 0.998 respectively.). Ion permeation rate (**e**) and single-ion selectivity (**f**) of TAPA-TFP-$x$-NH$_2$-CuBDC CMOF composite membranes (COFs: 0.5 mM TAPA + 0.5 mM TFP, $x$ = 0.25, 0.5, 1, 5). **g** Comparison of the K$^+$ permeation rate and ideal selectivity among membranes with various channel configurations. Information on the data points is given in Supplementary Table 6. **h** Normalized K$^+$ permeation rate and K$^+$/Na$^+$ selectivity of TAPA-TFP-0.25-NH$_2$-CuBDC CMOF composite membrane (COFs: 1 mM TAPA + 1 mM TFP) under different concentration gradients in the single-ion system. **i** Radar plot comparing the physicochemical properties and ion-sieving performance of TAPA-TFP-0.25-NH$_2$-CuBDC CMOF (COFs: 1 mM TAPA + 1 mM TFP) and 0.25-NH$_2$-CuBDC-TAPA-TFP MCOF composite membranes.

MOFs and COFs (Supplementary Fig. 14b, Supplementary Figs. 29–32, Supplementary Table 4). The top surface of the Synchronous$_{COF+MOF}$ composite membrane also presents a crater-like structure and definite MOFs are not observed, indicating that the 1D MOFs easily creep into the COF skeleton and disappear unless the MOFs are synthesized beforehand (Supplementary Fig. 33).

## K$^+$-selective transport property and mechanism

The cation transport properties in as-prepared composite membranes are investigated using a concentration-driven configuration (Fig. 4, Supplementary Fig. 34). For the CMOF composite membrane, the ion conductivity of the permeate side increases linearly with time, indicating a concentration-driven diffusion process (Fig. 4a, d,

Supplementary Fig. 35). Unprecedented $K^+$-selective transport characteristics are displayed by the TAPA-TFP-0.25-$NH_2$-CuBDC CMOF composite membrane (COFs: 1 mM TAPA + 1 mM TFP), and the ion permeation rate strictly depends on the hydrated ionic diameter, presenting a sharp size cutoff of 6.62 Å based on the pore diameter of 6.8 Å (Figs. 2e and 4b, Supplementary Table 5). Definitely, when the hydrated ionic diameter (7.16 Å for $Na^+$, 7.64 Å for $Li^+$, and 8.56 Å for $Mg^{2+}$) is higher than 6.8 Å, the ion permeation rate is less than 0.60 mmol $m^{-2}$ $h^{-1}$, which is at least 80-time lower than that of $K^+$ with a hydrated ionic diameter of 6.62 Å, thus, featuring extremely high $K^+$/$Na^+$ (82.52) and $K^+$/$Mg^{2+}$ (1131.07) ideal selectivities (Fig. 4c), significantly higher than those of previously reported membranes with various channel configurations[26–39] (Fig. 4g). After a long period of testing, the membrane still maintains excellent integrity, toughness, and $K^+$-selective transport property (Supplementary Fig. 36). It is noteworthy that the growth and coordination of the 1D MOFs in the 1D channels of 2D COF membranes is intricate and dependent not only on the MOF ligand concentration but also on the thickness of the pristine membrane, resulting in significant variations in ion selectivity between different CMOF membranes (Fig. 4a–f). The ion permeation rate of the TAPA-TFP-5-$NH_2$-CuBDC CMOF composite membrane (COFs: 1 mM TAPA + 1 mM TFP) is higher than that of the pristine COF membrane, suggesting that the high ligand concentration may cause defects in the composite membrane, leading to nonselective ion transports. When selecting the thinner pristine COF membrane, the ion selectivity of the as-prepared CMOF membrane decreases to a certain extent, which may be attributed to the shorter ion-channel wall interaction path (Fig. 4e, f). To evaluate the effect of driving force for ion diffusion, different concentration gradients including 0.02, 0.1, and 0.2 M are employed for $K^+$ and $Na^+$ transport. The $K^+$ permeation rate of TAPA-TFP-0.25-$NH_2$-CuBDC CMOF composite membrane (COFs: 1 mM TAPA + 1 mM TFP) presents a proportional rise to the concentration gradient, but $K^+$/$Na^+$ selectivity may show a nearly 10-fold decrease if the concentration gradient is low (Fig. 4h). The CMOF composite membrane is also employed for separating the practical brine system from Sichuan deep ground (pH = 6.74, $K^+$: 41617 ppm, $Na^+$: 275339 ppm, $Li^+$: 1553 ppm, $Mg^{2+}$: 9853 ppm, and other unknown components in unknown concentrations), and after 60 h of diffusion, the $K^+$ mass content increases by ~2 times, which demonstrates the preferential transport of $K^+$ than $Na^+$ (Supplementary Fig. 37). The not entirely satisfactory actual brine separation performance is attributed to the complex brine composition and intricate transport process due to numerous unknown factors affecting diffusion. Compared with CMOF composite membranes, MCOF composite membranes have larger pore size, surface area and MOF loadings, leading to higher $K^+$ permeation rate but poor ion-sieving performance (Fig. 4i, Supplementary Fig. 38). In binary-ion systems, the ion selectivity decreases due to the competition for occupying effective mass transport channels between cations (Supplementary Fig. 39)[26,27]. Thin-film composite (TFC) membranes were fabricated on anodic alumina oxide (AAO) substrates via in-situ confinement and growth (Supplementary Fig. 40) to investigate the practical application prospects. Due to the different nano-confined growth environment, the TFC membranes exhibit different properties from self-standing membranes, but still have $K^+$-recognition channels (Supplementary Figs. 41–44).

The CMOF membrane exhibits two different-sized pores (Fig. 2e). The 0.68-nanometer pores are dedicated to effective ion screening, while the 1.2-nanometer pores facilitate rapid ion migration (Fig. 5a). Pore-entrance size-sieving effect plays a significant role in the ultra-selective $K^+$ transport. The hydration energies of $Na^+$ ($-365$ kJ $mol^{-1}$), $Li^+$ ($-475$ kJ $mol^{-1}$), and $Mg^{2+}$ ($-1830$ kJ $mol^{-1}$) are much larger than that of $K^+$ ($-295$ kJ $mol^{-1}$), resulting in a higher transport energy barrier to entry into the narrow hetero-channels, and therefore come with exponentially lower permeation rates[29]. In addition to the pore-entrance sieving effect, in-pore transport dynamics also play vital roles

in ion sieving. In this stage, specific interactions (hydrogen bonding interactions, ion-binding site interactions) are often considered as the energy barriers for ion transport. The 1D MOFs have a large number of free -COOH groups with lower affinity for $K^+$ (Fig. 5b, c), which is conducive to the fast transportation of $K^+$ in the heterogeneous channels[40,41]. Molecular dynamics (MD) simulations also confirm the selective transport of $K^+$ ions over other metal ions (especially $Na^+$ ions) in the CMOF channels, which is supported by the snapshots showing the transmembrane migration behavior of $Na^+$ (yellow) and $K^+$ ions (pink) through the CMOF membranes (Fig. 5d), and the ion transmembrane energy barrier (Fig. 5e). As displayed in Fig. 5e, Potentials of mean force (PMF) profiles exhibit a gradual increase in energy as all ions approach the membrane pores, and $K^+$ ions face the lowest energy barriers during transmembrane migration, thereby facilitating $K^+$ ion transport and rapid transmembrane (The transmembrane energy barrier for $Mg^{2+}$ is the highest (34.51 kcal $mol^{-1}$), which is much greater than that of $Li^+$ (18.73 kcal $mol^{-1}$), $Na^+$ (15.56 kcal $mol^{-1}$), and $K^+$ (8.12 kcal $mol^{-1}$)). Furthermore, during the same period, the number of $K^+$ ions passing through the CMOF membrane is significantly higher than that of $Na^+$ ions, indicating superior $K^+$/$Na^+$ ideal selectivity (Fig. 5f), which is consistent with experimental results (confirming the efficacy of CMOF membranes in ion separation and lower actual selectivity). These findings suggest that the excellent permselectivity of the CMOF membrane is attributed to the synergistic effect of pore-entrance size-sieving and in-pore transport dynamics.

### Versatility
Additional MCOF and CMOF composite membranes have also been prepared according to the synthetic procedure. The pristine DABA-TFP COF membrane features a thickness of 70.21 nm. After the introduction of MOFs, its thickness increases to 116.33 nm (Supplementary Fig. 45). Although no explicit MOFs are observed, EDXS confirms a uniform distribution of MOFs in the CMOF composite membrane (Supplementary Fig. 45). Conversely, the pristine TAPA-DHA COF membrane features a thickness of 102.56 nm, but the $NH_2$-CuBDC-TAPA-DHA MCOF composite membrane presents a thickness of 61.54 nm (Supplementary Fig. 46). MOF lamellae are mainly present on the top surface of the MCOF composite membrane, and EDXS confirms a uniform distribution of MOFs, but exhibiting no obvious ion selectivity (Supplementary Figs. 46, 47).

### Discussion
In conclusion, in this work, a series of 1D MOF-in-2D COF hetero-structured composite membranes are prepared by bottom-up strategy. Induced by the coordination interaction between the -NH groups in COFs and the Cu centers from MOFs, the composite membranes present a molecular-level interlinked hybridization of covalent and metal organic hetero-frameworks. The resulting CMOF composite membrane with biomimetic KcsA channels allows for precise $K^+$-recognition transport. Pore-entrance sieving effect and channel wall-ion interactions enable ultra-selective and fast transport of $K^+$ with unprecedented $K^+$/$Na^+$ selectivity of 82.52 and $K^+$/$Mg^{2+}$ selectivity of 1131.07. This work sheds light on designing and developing single-species selective artificial membranes for precious ion recovery.

### Methods
#### Materials
1,3,5-triformylphloroglucinol (TFP, 97%), tris (4-aminophenyl) amine (TAPA, 98%), 2, 5-diaminobenzenesulfonic acid (DABA, 98%), and 2,5-dihydroxyterephthalaldehyde (DHA, 97%) were ordered from Shanghai Tensus Biotech Co. Ltd. (Shanghai, China) and used without any further purification. $Cu(NO_3)_2$ (analytical grade purity) and 2-aminoterephthalic acid ($NH_2$-BDC, 98%) were purchased from Aladdin (Shanghai, China). Potassium chloride (KCl, analytical grade purity), sodium chloride (NaCl, analytical grade purity), lithium chloride

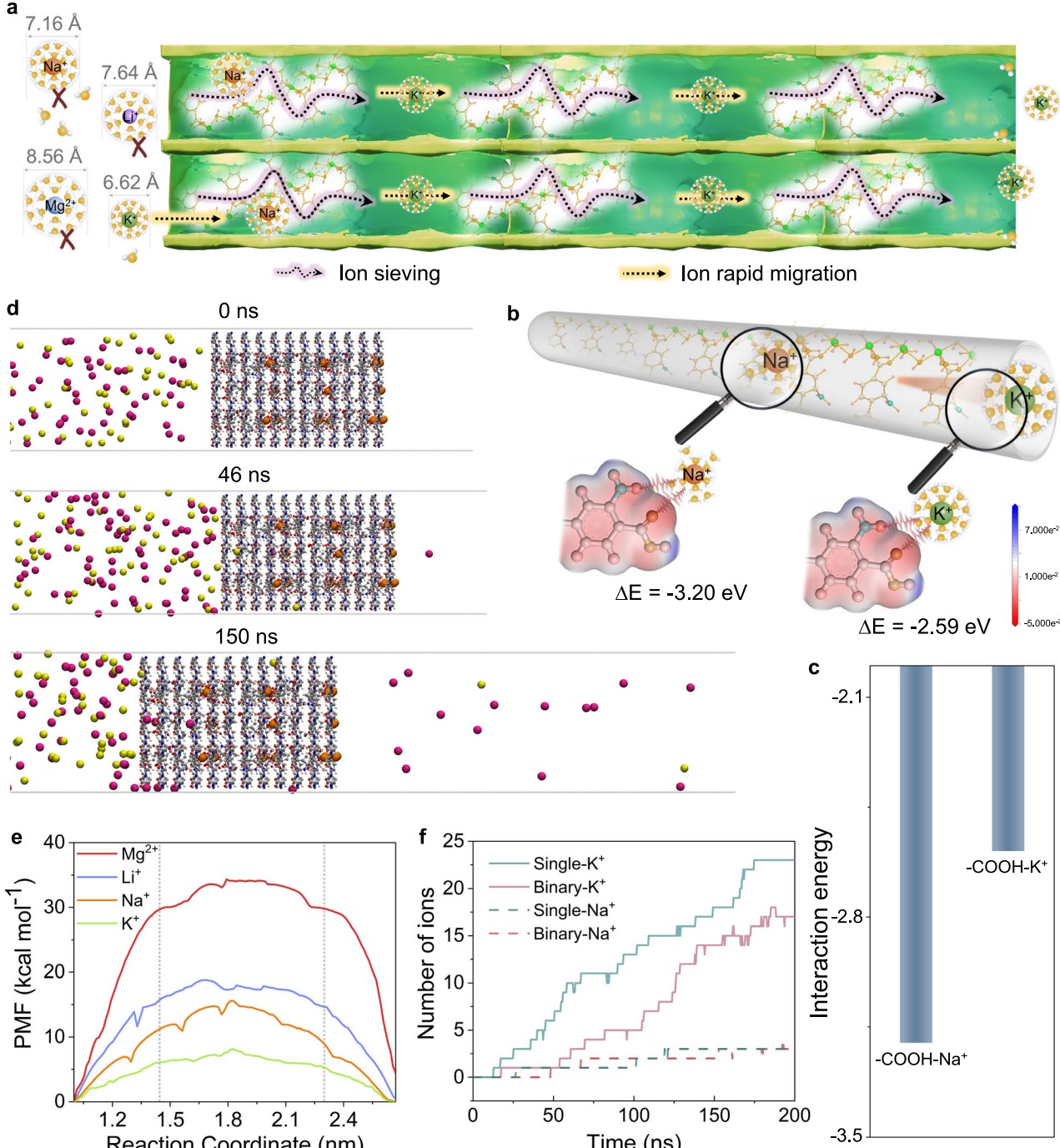

**Fig. 5 | K⁺-selective transport mechanism. a** Schematic diagram of ion transport process. The interaction energy (**c**) and schematic diagram (**b**) obtained from density functional theory (DFT) calculations. **d** Simulation snapshots visually depict the transmembrane behavior of Na⁺ (yellow) and K⁺ ions (pink) (C, O, N, H, and Cu are depicted in gray, red, blue, white and orange respectively). **e** PMF profiles for K⁺, Na⁺, Li⁺ and Mg²⁺ ions across the CMOF membrane. **f** Transmembrane ion permeation kinetics across the CMOF membrane in single-ion and binary-ion systems.

(LiCl, analytical grade purity), magnesium chloride (MgCl₂, analytical grade purity), dichloromethane (DCM, analytical grade purity), and methanol (analytical grade purity) were ordered from Sinopharm Chemical Reagent Co. Ltd. (Shanghai, China) and used as received. Petri dishes and glass sheets were purchased from Shanghai Glass Instrument Factory (Shanghai, China).

## Interfacial synthesis of the CMOF composite membrane
The TAPA-TFP-x-NH₂-CuBDC CMOF membranes were synthesized at room temperature and atmospheric pressure by oil-water interfacial polymerization. First, TAPA-TFP self-standing COF membranes were fabricated by oil-water interfacial polymerization[18]. After the COF membrane was successfully prepared at the interface, NH₂-BDC dissolved in DCM (NH₂-BDC was first dissolved with DMF and then dissolved in DCM) was slowly injected into the underlying oil phase by an injection syringe (The syringe needle tip is perpendicular to the liquid surface with the injection rate of 1 mL min⁻¹). One day later, Cu(NO₃)₂ aqueous solution was slowly injected into the upper aqueous phase (The syringe needle tip is perpendicular to the liquid surface with the injection rate of 0.5 mL min⁻¹). It should be noted that, if the injection

rates of $NH_2$-BDC and $Cu(NO_3)_2$ are too high, it may generate excessive bubbles and damage the integrity of the COF membrane at the interface, which will significantly affect the morphology and separation performance of the CMOF membrane. Due to its small hydrated diameter (0.8 nm) and capillary effect, $Cu^{2+}$ could easily drill into the channels and coordinate with $NH_2$-BDC. Three days later, TAPA-TFP-$x$-$NH_2$-CuBDC CMOF membranes can be observed at the DCM-water interface. Before further characterization, the membranes were fully soaked and washed by methanol and water. The synthesis of DABA-TFP-$NH_2$-CuBDC CMOF membranes is similar to the above process (DABA: 3 mM, TFP: 2 mM, $NH_2$-BDC: 0.5 mM, $Cu(NO_3)_2$: 10 mM). Furthermore, the synthesis process of TAPA-TFP-$x$-$dd$-$NH_2$-CuBDC CMOF membranes is similar to that of TAPA-TFP-$x$-$NH_2$-CuBDC CMOF membranes, while both the $NH_2$-BDC and $Cu(NO_3)_2$ solutions ($Cu(NO_3)_2$ was first dissolved with DMF and then dissolved in DCM) were slowly injected into the underlying oil phase.

### Interfacial synthesis of the MCOF composite membrane

The $x$-$NH_2$-CuBDC-TAPA-TFP MCOF membranes were synthesized at room temperature and atmospheric pressure by oil-water interfacial polymerization. First, $x$-$NH_2$-CuBDC MOFs were prepared at the oil-water interface[19]. Subsequently, TFP (2 mM) dissolved in DCM was slowly injected into the underlying oil phase by an injection syringe to allow the -$NH_2$ of $NH_2$-BDC to fully react with the -CHO of TFP (This movement should be as slow and gentle as possible to avoid interface disturbance). One day later, TAPA (2 mM) dissolved in DCM was slowly injected into the underlying oil phase to initiate the Schiff base reaction for COF growth. In the whole process, the operation should be gentler than during the preparation of CMOF membranes because the MOFs at the interface is merely loosely assembled through van der Waals forces rather than forming a resilient membrane, making the MOFs more prone to random migration, which will ultimately compromise the quality of MCOF membranes. The syringe needle tip is perpendicular to the liquid surface with the injection rate of 0.3 mL min⁻¹ when introducing TAPA and TFP. Three days later, $x$-$NH_2$-CuBDC-TAPA-TFP MCOF composite membranes can be observed at the DCM-water interface. Before further characterization, the membranes were fully soaked and washed by methanol and water. The synthesis of $NH_2$-CuBDC-TAPA-DHA MCOF membranes is similar to the above process (TAPA: 1 mM, DHA: 1.5 mM, $NH_2$-BDC: 0.5 mM, $Cu(NO_3)_2$: 10 mM).

### Interfacial synthesis of the synchronous$_{COF+MOF}$ membrane

The Synchronous$_{COF+MOF}$ membranes were synthesized at room temperature and atmospheric pressure by oil-water interfacial polymerization. Typically, TAPA (2 mM), TFP (2 mM) and $NH_2$-BDC (0.5 mM) were respectively dissolved with DCM to form oil phase I, II and III. Subsequently, equal volumes of oil phase I, II and III were poured into a petri dish. After standing for 3 min, $Cu(NO_3)_2$ aqueous solution (10 mM) was gently added onto the top surface of the oil phase, keeping a peaceful and stable oil/water interface. Three days later, Synchronous$_{COF+MOF}$ membranes can be observed at the DCM-water interface. Before further characterization, the membranes were fully soaked and washed by methanol and water.

### Characterizations

Chemical structures were characterized by a Bruker TENSOR II Fourier transform infrared spectrometer (FTIR) with a spectral range of 400-4000 cm⁻¹, a Nicolet iN10 attenuated total reflection Fourier transform infrared microscope (ATR, Thermo Scientific), and an X-ray photoelectron spectroscopy (XPS, Kratos AXIS SUPRA+) equipped with Al Kα X-ray source (1486.6 eV). Powder X-ray diffraction (PXRD, Bruker D8 Advance) was employed in a 2θ range of 3–50° utilizing Cu K-α (λ = 1.54 Å) as the X-ray radiation source. Grazing incidence wide angle X-ray scattering (GIWAXS) measurements were conducted on a small/ wide angle X-ray scattering system (Xenocs Xeuss 2.0) equipped with a Pilatus 3R 300 K 2D detector and the membrane sample was irradiated at a grazing angle of 0.2°. The morphology was investigated by a field emission scanning electron microscope (SEM, GeminiSEM 500) and before measurements, samples were sprayed with Pt/Pd alloy for 40 s using an auto fine coater (208 HR, TED PELLA, INC). JEM-2100F field emission transmission electron microscope was used to further analyze the morphology and crystallinity. Thermal stability was evaluated by a thermogravimetric analyzer (DTG-60H, SHIMADZU) in the temperature range of 25–800 °C with a heating ramp of 5 °C min⁻¹ under a flowing $N_2$ atmosphere (50 mL min⁻¹). A contact angle measuring instrument (POWEREACH, Shanghai zhongchen digital technic apparatus co. ltd) was used to measure the water contact angles (WCAs) in air. Pore size distribution and surface area were characterized by an automatic microporous physical and chemical adsorption instrument (ASAP2020M + C, Micromeritics, USA) and examined by measuring the $N_2$ adsorption-desorption isotherm at 77 K in liquid nitrogen bath. Inductively Coupled Plasma-Optical Emission Spectrometry (ICP-OES, Optima 7300 DV, USA) was employed to detect the ion concentration. A conductivity meter (FE38, Mettler Toledo) was used to determine the ion conductivity. The UV-Vis-NIR spectra were obtained by a UV-Vis-NIR spectrophotometer (SolidSpec-3700i DUV, Shimadzu) equipped with a mercury lamp. The Cu $L$-edge synchrotron-based X-ray absorption near edge structure spectroscopy were measured on beamline BL12B of the National Synchrotron Radiation Laboratory, Hefei, using the total electron yield mode by collecting the sample drain current under a chamber pressure of <10⁻⁷ Pa. The energy range was 915–965 eV with an energy resolution of ca. 0.2 eV. The tensile properties were measured by an electronic tensile testing machine (DEBEN5000DL) with a tensile rate of 0.1 mm min⁻¹ controlled by a microcomputer (In an aqueous solution, a circular membrane formed in the petri dish is folded twice (first in half, then again in half), resulting in a composite membrane with four layers tightly stacked together. Subsequently, the composite membrane is carefully lifted out from the aqueous solution using a substrate and allowed to air-dry naturally. The dried membrane is then trimmed into 40 mm × 10 mm strips for mechanical tensile testing.).

### Ion permeation experiment

A homemade permeation cell was employed to investigate the single and binary ion permeation behaviors. The valid membrane test area is 2.27 cm². In a typical ion permeation experiment, 50 mL of salt solution is poured into the feed side, and 50 mL of DI water is poured into the permeate side. Both the feed and permeate sides are magnetically stirred to avoid the concentration polarization. The ion conductivity and ion concentration in the permeate side are respectively determined by a conductivity meter and an ICP-OES. In single ion permeation experiments, 0.1 M KCl, 0.1 M NaCl, 0.1 M LiCl or 0.1 M $MgCl_2$ aqueous solutions were used as the feed solutions. In binary ion permeation experiments, 0.1 M KCl/0.1 M NaCl, 0.1 M KCl/0.1 M $MgCl_2$, or 0.1 M LiCl/0.1 M $MgCl_2$ were employed as the separation system. Three replicates were conducted for each salt solution.

### DFT calculations

DFT calculations were carried out by the Gaussian 16 software. The B³LYP functional was adopted for all calculations in combination with the D3BJ dispersion correction. In geometry optimization and frequency calculations, the 6–31 G(d,p) basis set was used.

### Molecular dynamics (MD) simulations

MD simulations were conducted using the GROMACS 2021.7 program, with subsequent analysis carried out using VMD 1.9.3. LJ parameters for atoms within the CMOFs were obtained from the UFF4MOF2, and the OPC3 model was used to describe water. The REPEAT charges of CMOFs were calculated using CP2K under the PBE-D3/TZV2P-

MOLOPT-GTH method. The CMOF membrane was positioned in the middle-right region of a simulation box with dimensions of 5.8 nm × 3.4 nm × 26 nm. A specific number of water molecules and ions were added to the system. For the simulation of a binary salt system comprising KCl and NaCl at a concentration of 1 M each, we included 6890 water molecules, 104 $K^+$ ions, and 104 $Na^+$ ions. $Cl^-$ ions were added to neutralize the overall charge of the system, with an additional 5673 water molecules introduced on the permeate side. The simulation protocol involved energy minimization using the steepest descent method, followed by 200 ps of pre-equilibration at 300 K. Subsequently, a 150 ns equilibrium molecular dynamics simulation run was performed in the canonical ensemble (NVT), with the system temperature controlled at 300 K using the Nose-Hoover method. Cut-off distances were set to 1.2 nm, and the PME method was utilized to compute long-range electrostatic interactions. Periodic boundary conditions (PBC) were applied in two dimensions to minimize edge effects. Potentials of mean force (PMF) profiles for ions traversing the membrane were determined using umbrella sampling. Thirty-two windows were evenly distributed along the z-axis, covering a range from 1.0 to 2.6 nm for the ions, with each window undergoing an 8.0 ns simulation run.

## Data availability

The authors declare that all the data supporting the findings of this study are available within the article (and Supplementary Information files). Additional data are available from the corresponding author upon request. Source data are provided with this paper.

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

## Acknowledgements

The authors would like to thank the financial support from the National Key Research and Development Program of China (2023YFB3811000, J.L.), the National Natural Science Foundation of China (22478371, J.L.), the Young Talent Program (GG2400007003, J.L.). This work was partially carried out at the Instrument Center for Physical Science, University of Science and Technology of China.

## Author contributions

Q.S. and J.L. conceived and designed the research. J.L. acquired funding and supervised research and project administration. Q.S., P.D., J.D. and A.Y. conducted the experiments. Q.S., P.D., J.D., A.Y., D.C., J.M., S.U.H., J.G. and J.L. analyzed the experimental data. Q.S. wrote the paper. All authors gave approval to the final version of the manuscript.

## Competing interests

The authors declare no competing interests.
