## [Transparent Peer Review file · Nature Communications]

Biomimetic KcsA channels enabled by 1D MOF-in-2D COF

Corresponding Author: Professor Jiangtao Liu

Version 0:

Reviewer comments:

Reviewer #1

(Remarks to the Author)

This manuscript reports the preparation of a MOF-in-COF structured membrane for ion separation. The proposed channel structure is intriguing, but the XRD spectra lack COF peaks, raising concerns about its actual structure. Additionally, the claimed 0.68 nm channel size from BET analysis conflicts with another distinct peak larger than 1 nm that is not addressed. The actual pore size in an aqueous environment may differ from BET values, making the pore size sieving effect less convincing. While the membrane preparation and characterization experiments are comprehensive, the ion separation experiments and discussions are relatively weak and do not thoroughly reflect mechanisms like sieving or ion binding, as well as the case of separating ion mixtures.

1. The abstract does not clearly present the research idea. Why is a MOF-in-COF structure necessary? How can this structure specifically address ion/ion separation challenges and enable fast K⁺ ion transport?
2. The introduction should be revised for better organization. The authors mention coordination first, then introduce COFs, without a clear logical relationship between them. It is also unclear whether coordination is used for ion separation or hybridization of MOF and COF. While several effects of their structure are highlighted, it remains unclear how confined effects, ligand trapping, and interlocking growth address the challenge of ion separation.
3. In line 114, regarding EDXS characterization, it only shows element distribution but cannot prove successful MOFs formation.
4. Please clarify how Fig. S5 demonstrates that Cu ions alone cannot coordinate with COFs.
5. There is another intense peak (pore width around 1.5 nm) shown in the pore size distribution of Fig. 2e; please explain this observation and how a 0.68 nm pore size was formed by the interlocking of MOF and COF structures—specifically which space in this structure accounts for such small pore sizes.
6. Please provide HRTEM images of pure MOFs and COFs as references.
7. Since it is possible that MOF growth might disrupt original COF stacking due to missing peaks in XRD data, how did the authors ensure an ideal 1D MOFs within 2D COFs structure?
8. Please explain why there is a tenfold decrease in K⁺/Na⁺ selectivity within binary systems.
9. While pore entrance energy barriers can be supported by membrane pore size data, please also provide evidence for intrapore energy barriers. Please specify transport channels within this MOFs-COFs composite so authors may identify functional groups responsible for affecting transport since carboxyl is not solely present in composites.
10. Control samples like pure COFs membranes should be investigated to demonstrate the role of MOFs.

Reviewer #2

(Remarks to the Author)

The manuscript reports a biomimetic KcsA channel based on a 1D MOF-in-2D COF hetero-structured composite

membranes for highly selective ion separation. The composite membranes are formed via interlocking and in situ immobilized growth, employing a bottom-up strategy. The composite membranes present a molecular-level interlinked hybridization of covalent and metal organic hetero-frameworks. The resulting CMOF composite membrane with biomimetic KcsA channels allows precise K⁺ recognition transport. Pore-entrance sieving effect and channel wall-ion interactions enable ultra-selective and fast transport of K⁺ with unprecedented K⁺/Na⁺ selectivity of 10² and K⁺/Mg²⁺ selectivity of 10³. I think this manuscript is suitable for publication in Nature Communication, provided that the authors consider the following comments:

1. The manuscript mentions that the 2D TAPA-TFP COF membranes have superior tenacity and flexibility, yet no mechanical property tests are reported. Relevant mechanical testing, such as tensile tests, is recommended to support these claims.
2. The description of the interfacial polymerization process is too brief. For example, during CMOF membrane preparation, key details such as the injection rate of NH₂-BDC and Cu(NO₃)₂, the injection angle, and the control of interfacial stability are not fully explained. These factors can significantly impact membrane homogeneity and performance. The authors should provide more detailed experimental procedures.
3. Further studies, including multiple repeat experiments and continuous operation over days or weeks, are recommended to demonstrate the stability of the membrane performance. Additionally, the membrane's stain resistance tests should be conducted to evaluate.
4. The manuscript explains the ion selective transport mechanism through experimental data and basic theoretical analysis. It is recommended that the authors incorporate molecular dynamics simulations to explore the ion transport paths, energy barriers, and interactions with the channel walls in the 1D MOF-in-2D COF channels.
5. More relevant membranes related works should be cited, such as Yang et al., Science 2019, 364, 1057-1062.; Wang et al., ACS Nano 2024, 18, 34698-34707.; Hong et al., Nature Commun. 2024, 15:3160.

Version 1:

Reviewer comments:

Reviewer #1

(Remarks to the Author)

The authors have well addressed my concerns in their previous manuscript.

Reviewer #2

(Remarks to the Author)

The authors have addressed all the questions I have raised, and this manuscript can be accepted as it is.

Response to Reviewer #1:

Comments:

This manuscript reports the preparation of a MOF-in-COF structured membrane for ion separation. The proposed channel structure is intriguing, but the XRD spectra lack COF peaks, raising concerns about its actual structure. Additionally, the claimed 0.68 nm channel size from BET analysis conflicts with another distinct peak larger than 1 nm that is not addressed. The actual pore size in an aqueous environment may differ from BET values, making the pore size sieving effect less convincing. While the membrane preparation and characterization experiments are comprehensive, the ion separation experiments and discussions are relatively weak and do not thoroughly reflect mechanisms like sieving or ion binding, as well as the case of separating ion mixtures.

Answer: Thank you very much for your professional suggestions. Indeed, the XRD spectra show MOF peaks but lack COF peaks. However, during the process of preparing the CMOF membrane, the ordered and definite structure of the COF membrane has been established and ensured before (as shown in the following figure). On this basis, the ligands and metal for forming MOF are successively introduced to construct the CMOF structure. The XRD characteristic peaks from COFs that should have been detected in the CMOF structure were not detected. The reasons maybe in two aspects as follows: (1) When the XRD peaks of MOF and COF coexist, the peak from MOF will significantly compress that of COF, making it impossible to detect. (2) The structure of MOF within the ordered COF pores masks the lattice signal of COF.

The ordered and definite structure of the COF membrane had already been ensured before forming the CMOF membrane. If we want to confirm the actual structure of CMOF, the most important step is to examine the state of MOF within the COF channels (the detected XRD signal from MOFs matters).

As shown in the figure below, the XRD signal of MOF is significantly stronger than that of COF. It should be noted that even after reducing the MOF signal intensity by fivefold when processing the data for plotting, it remains stronger than the XRD signal of COF. This means that when MOF and COF coexist, the intense XRD signal from MOF will overwhelmingly suppress the COF signals, making the XRD peaks of COF extremely difficult to observe. Furthermore, as shown in the figure below, the (110) crystallographic plane of the COF can be observed, but the signal is weaker, which may be attributed to the introduction of the MOF reducing the crystallinity and masking the (100) crystal plane (making it difficult to be detected) of COF. Meanwhile, the right-shift of the (110) crystallographic plane confirms the effective growth of MOF within the COF channels and the relatively strong interaction between MOF and COF. This discussion has been added to the revised manuscript.

Original: It is worth mentioning that the sharp XRD peaks of the pristine COF membrane are not distinctly observed in the CMOF composite membrane, which may result from the introduction of MOFs into the pore structure or interlayer of COFs based on coordination interactions disrupting the long-range periodicity of the porous structure of COFs^{17,23}.

Revised: It is worth mentioning that the sharp XRD peaks of the pristine COF membrane that should have been detected (The ordered and definite structure of the COF membrane had already been established before forming the CMOF membrane as displayed in **Figure 2a**.) are not distinctly observed in the CMOF composite membrane, which may result from: 1) the introduction of MOFs into the pore structure or interlayer of COFs based on coordination interactions masking the XRD signals from COFs, and 2) the intense XRD signals from MOFs overwhelmingly suppressing the COF signals, making the XRD peaks from COFs extremely difficult to observe when MOFs and COFs coexist^{17,23}. Still, the (110) crystallographic plane with weak signals of pristine COF membrane can be observed in CMOF composite membrane (**Figure 2f**), and its right-shift compared with pristine COF membrane confirms the effective growth of MOFs within the COF channels and the relatively strong interaction between MOFs and COFs.

Furthermore, to address this issue, we conducted TEM characterization and identified distinct crystalline domains and lattice fringes corresponding to COF and MOF respectively, thereby confirming their coexistence (as shown in the figure below).

In addition, to clarify the actual structure of 1D MOF-in-2D COF as closely as possible, we conducted structural simulations to visualize the 1D MOF-in-2D COF hetero-structure based on the characterization results. As shown in the following figure, the distance between Cu center from MOFs and its neighboring N atom from COFs ranges from 0.28 to 0.73 nm, which confirms the short-range coordination interaction between MOFs and COFs. This discussion has been added to the revised manuscript.

Figure 1. (f) The simulated 1D MOF-in-2D COF hetero-structure (Some atoms are omitted for clarity.). The distance between Cu center from MOFs and its neighboring N atom from COFs ranges from 0.28 to 0.73 nm, which confirms the short-range coordination interaction between MOFs and COFs.

Added: Simulation results indicate the short-range coordination interaction between MOFs and COFs (Supplementary Figure 1f).

In addition to the 0.68 nm channel size from BET analysis, another peak is observed, for which we have provided a detailed explanation regarding its emergence in the revised manuscript (as shown in the figure below).

Added: It is noteworthy that the CMOF composite membrane exhibits another weaker peak at 1.20 nm, which is similar to the pore size of pristine COF membrane. This suggests that the chain-like MOFs may not fully occupy the COF pore channels in a complete top-down filling manner (Supplementary Figure 7f).

Figure 7. Schematic diagram of CMOF pore channels (f).

We also conducted simulations on the pore diameters of CMOF and COF. It was found that the pore diameter of COF membrane is approximately 1.17 nm. After introducing MOFs, the pore diameter decreases to 0.67 nm, which is basically consistent with the experimental characterization results. This discussion has been added to the revised manuscript.

Figure 7. Simulated pore size distributions of pristine COFs (c) and 1D MOF-in-2D COF hetero-structure (derived from Figure 7e) (d). The simulated 1D MOF-in-2D COF hetero-structure (Some atoms are omitted for clarity.) (e).

Original: The TAPA-TFP-0.25-NH₂-CuBDC CMOF composite membrane features a main pore diameter of 0.68 nm, which is smaller than that of COFs (1.22 nm) or MOFs (1.42 nm),.....

Revised: The TAPA-TFP-0.25-NH₂-CuBDC CMOF composite membrane features a main pore diameter of 0.68 nm (basically consistent with simulation results), which is smaller than that of COFs (1.22 nm) or MOFs (1.42 nm),.....

We totally agree with you that the actual pore size in an aqueous environment may differ from BET values due to the presence of charged carboxyl groups in the channel. To figure out this matter, we conducted simulation calculations (RDF). The computation results suggest that, the RDF peak value is relatively low (as shown in the left part of the figure below), indicating that the probability of water molecules appearing around -COOH groups in CMOF channels is low. Therefore, a stable hydration layer cannot be formed. It is different from the previously reported literature (Without the subnanometer confinement effect, the hydration behavior is not restricted.) (as shown in the right part of the figure below). Consequently, in the complex confined channels with a diameter of 0.68 nm, although the carboxyl groups carry significant charges, they hardly form hydration layers. Therefore, the aperture obtained from BET can represent the effective aperture for ion transport. This discussion has been added to the revised manuscript.

Figure 7. Radial distribution function (RDF) of oxygen in water molecules around -COOH groups in CMOF channels (The RDF peak value is relatively low, indicating that the probability of water molecules appearing around -COOH groups in CMOF channels is low. Therefore, a stable hydration layer cannot be formed.) (g).

Added: Computation results indicate that in the complex confined channels with a diameter of 0.68 nm, although the -COOH groups carry significant charges in an aqueous solution environment, they are almost unable to combine water molecules to form stable hydrated layers. Therefore, the pore size obtained from BET can represent the effective aperture for ion transport (**Supplementary Figure 7g**).

Following your suggestions, in addition to the pore size sieving effect, ion transport dynamics within the channels have also been investigated using MD simulations. The results and discussion have been added to the revised manuscript.

Figure 5 | K⁺-selective transport mechanism. **a**, Schematic diagram of ion transport process. **b-c**, The interaction energy obtained from density functional theory (DFT) calculations. **d**, Simulation snapshots visually depict the transmembrane behavior of Na⁺ (yellow) and K⁺ ions (pink) (C, O, N, H, and Cu are depicted in grey, red, blue, white and orange respectively.). **e**, PMF profiles for K⁺, Na⁺, Li⁺ and Mg²⁺ ions across the CMOF membrane. **f**, Transmembrane ion permeation kinetics across the CMOF membrane in single-ion and binary-ion systems.

Added: The CMOF membrane exhibits two different-sized pores (**Figure 2e**). The 0.68-nanometer pores are dedicated to effective ion screening, while the 1.2-nanometer pores facilitate rapid ion migration (**Figure 5a**).

Added: Molecular dynamics (MD) simulations also confirm the selective transport of K⁺ ions over other metal ions (especially Na⁺ ions) in the CMOF channels, which is supported by the snapshots showing the transmembrane migration behavior of Na⁺ (yellow) and K⁺ ions (pink) through the CMOF membranes (**Figure 5d**), and the ion transmembrane energy barrier (**Figure 5e**). As displayed in **Figure 5e**, Potentials of mean force (PMF) profiles exhibit a gradual increase in energy as all ions approach the membrane pores, and K⁺ ions face the

lowest energy barriers during transmembrane migration, thereby facilitating K^+ ion transport and rapid transmembrane (The transmembrane energy barrier for Mg^{2+} is the highest ($34.51 \text{ kcal mol}^{-1}$), which is much greater than that of Li^+ ($18.73 \text{ kcal mol}^{-1}$), Na^+ ($15.56 \text{ kcal mol}^{-1}$), and K^+ ($8.12 \text{ kcal mol}^{-1}$)). Furthermore, during the same period, the number of K^+ ions passing through the CMOF membrane is significantly higher than that of Na^+ ions, indicating superior K^+/Na^+ ideal selectivity (**Figure 5f**), which is consistent with experimental results (confirming the efficacy of CMOF membranes in ion separation and lower actual selectivity). These findings suggest that the excellent permselectivity of the CMOF membrane is attributed to the synergistic effect of pore-entrance size-sieving and in-pore transport dynamics.

Added: Molecular dynamics (MD) simulations. MD simulations were conducted using the GROMACS 2021.7 program, with subsequent analysis carried out using VMD 1.9.3. LJ parameters for atoms within the CMOFs were obtained from the UFF4MOF2, and the OPC3 model was used to describe water. The REPEAT charges of CMOFs were calculated using CP2K under the PBE-D3/TZV2P-MOLOPT-GTH method. The CMOF membrane was positioned in the middle-right region of a simulation box with dimensions of $5.8 \text{ nm} \times 3.4 \text{ nm} \times 26 \text{ nm}$. A specific number of water molecules and ions were added to the system. For the simulation of a binary salt system comprising KCl and NaCl at a concentration of 1 M each, we included 6890 water molecules, 104 K^+ ions, and 104 Na^+ ions. Cl^- ions were added to neutralize the overall charge of the system, with an additional 5673 water molecules introduced on the permeate side. The simulation protocol involved energy minimization using the steepest descent method, followed by 200 ps of pre-equilibration at 300 K. Subsequently, a 150 ns equilibrium molecular dynamics simulation run was performed in the canonical ensemble (NVT), with the system temperature controlled at 300 K using the Nose-Hoover method. Cut-off distances were set to 1.2 nm, and the PME method was utilized to compute long-range electrostatic interactions. Periodic boundary conditions (PBC) were applied in two dimensions to minimize edge effects. Potentials of mean force (PMF) profiles for ions traversing the membrane were determined using umbrella sampling. Thirty-two windows were evenly distributed along the z-axis, covering a range from 1.0 to 2.6 nm for the ions, with each window undergoing an 8.0 ns simulation run.

1. The abstract does not clearly present the research idea. Why is a MOF-in-COF structure necessary? How can this structure specifically address ion/ion separation challenges and enable fast K^+ ion transport?
Answer: Thank you very much for your professional suggestions. Following your suggestion, in the revised manuscript, we have rephrased the abstract to clearly present the research idea, including “why is a MOF-in-COF structure necessary? How can this structure specifically address ion/ion separation challenges and enable fast K^+ ion transport”.

Added: Considering the non-homogeneous heterostructure of KcsA channels and -COOH groups generally showing lower K^+ affinity, we propose the "1D MOF (rich in -COOH groups)-in-2D COF" concept, aiming to enhance K^+/Na^+ separation through strategic construction of heterogeneous ion transport channels, therefore narrowing the pore size of pristine COF membrane, and weakening the K^+ -channel wall interactions.

2. The introduction should be revised for better organization. The authors mention coordination first, then introduce COFs, without a clear logical relationship between them. It is also unclear whether coordination is used for ion separation or hybridization of MOF and COF. While several effects of their structure are highlighted, it remains unclear how confined effects, ligand trapping, and interlocking growth address the challenge of ion separation.

Answer: Thank you very much for your professional suggestions. Following your suggestion, in the revised manuscript, the introduction has been revised for better organization: we first proposed the separation mechanism of KcsA and highlighted the importance of channel wall-ion interactions in ion separation. We then outlined several biomimetic membranes based on the mechanism of channel wall-ion interactions. Subsequently, we pointed out the limitations of these membrane designs and proposed that heterogeneous structures might enable higher selectivity. We then discussed the advantages of using

COFs for constructing heterogeneous composite membranes and provided examples. Subsequently, we pointed out the drawbacks of the exemplified 3D MOF in COF structures and introduce the advantages of 1D MOF-in-2D COF of this work.

To minimize misunderstandings, the description of the coordination for ion separation is removed.

For the confined effects, ligand trapping, and interlocking growth, they merely concern the construction and growth process of the 1D MOF-in-2D COF structure, emphasizing the process and the membrane growth mechanism. The final heterogeneous structure showing lower K^+ affinity is used to address the challenge of ion separation.

3. In line 114, regarding EDXS characterization, it only shows element distribution but cannot prove successful MOFs formation.

Answer: Thank you very much for your valuable suggestions. We totally agree with you that EDXS characterization only shows element distribution. However, the detected copper signal from EDXS of CMOF or MCOF membranes almost exclusively originate from MOFs, as isolated copper ions from $Cu(NO_3)_2$ neither can coordinate with COFs nor exist within it, and thus cannot be detected (has been confirmed, as shown in the following figure). Therefore, EDXS characterization can indirectly prove successful MOFs formation.

Figure 5. SEM images and EDXS mapping of the prepared membrane referring to the synthesis method of CMOF membranes, except that NH_2 -BDC is not added to the underlying oil phase (COFs: 2 mM TAPA + 2 mM TFP). Based on the SEM images and EDXS mapping, it is concluded that isolated copper ions from $Cu(NO_3)_2$ cannot coordinate with COFs. The reasons are as follows in two aspects: (1) No copper signal was detected on the membrane ($COF+Cu(NO_3)_2$). If isolated copper ions could coordinate with COFs, their signal should theoretically have been detectable by EDXS mapping (Note that EDXS mapping is sensitive to the concentration of the element being detected, and herein the thicker pristine membrane that may trap more copper ions was chosen in order to maximize the detection of the copper ion signal). (2) The long vermicular nanotubes distributed on the bottom surface of the prepared membrane remained in a plump state. If isolated copper ions could coordinate with COFs, the generated heat would have caused the plump COFs to become shrunken or even broken (like the nanotubes distributed on CMOF or MCOF membranes in Figure 2 or Figure 17).

4. Please clarify how Fig. S5 demonstrates that Cu ions alone cannot coordinate with COFs.

Answer: Thank you very much for your valuable suggestions. Following your suggestion, in the revised manuscript, we have clarified how Fig. S5 demonstrates that Cu ions alone cannot coordinate with COFs.

Original: **Figure 5.** SEM images and EDXS mapping of the prepared membrane referring to the synthesis method of CMOF membranes, except that NH₂-BDC is not added to the underlying oil phase (COFs: 2 mM TAPA + 2 mM TFP). Cu signals are not observed on the membrane and the long vermicular nanotubes distributed on the bottom surface of the prepared membrane are plump. Note that EDXS mapping is sensitive to the concentration of the element being detected, and herein the thicker pristine membrane that may trap more copper ions was chosen in order to maximize the detection of the copper ion signal.

Revised: **Figure 5.** SEM images and EDXS mapping of the prepared membrane referring to the synthesis method of CMOF membranes, except that NH₂-BDC is not added to the underlying oil phase (COFs: 2 mM TAPA + 2 mM TFP). Based on the SEM images and EDXS mapping, it is concluded that isolated copper ions from Cu(NO₃)₂ cannot coordinate with COFs. The reasons are as follows in two aspects: (1) No copper signal was detected on the membrane (COF+Cu(NO₃)₂). If isolated copper ions could coordinate with COFs, their signal should theoretically have been detectable by EDXS mapping (Note that EDXS mapping is sensitive to the concentration of the element being detected, and herein the thicker pristine membrane that may trap more copper ions was chosen in order to maximize the detection of the copper ion signal.). (2) The long vermicular nanotubes distributed on the bottom surface of the prepared membrane remained in a plump state. If isolated copper ions could coordinate with COFs, the generated heat would have caused the plump COFs to become shrunken or even broken (like the nanotubes distributed on CMOF or MCOF membranes in **Figure 2** or **Figure 17**).

Additionally, numerous studies have demonstrated that the imine sites in COF structures can coordinate with copper ions from copper acetate (Energy Environ. Mater. 2024, 7, e12732, Nat. Commun. 2022, 13, 5768), while there have been hardly any reports of coordination with copper ions from copper nitrate.

5. There is another intense peak (pore width around 1.5 nm) shown in the pore size distribution of Fig. 2e; please explain this observation and how a 0.68 nm pore size was formed by the interlocking of MOF and COF structures—specifically which space in this structure accounts for such small pore sizes.

Answer: Thank you very much for your professional suggestions. Following your suggestion, in the revised manuscript, we have explained this observation and how a 0.68 nm pore size was formed-

specifically which space in this structure accounts for such small pore sizes (as shown in the following figure).

Figure 7. Schematic diagram of CMOF pore channels (f).

Added: It is noteworthy that the CMOF composite membrane exhibits another weaker peak at 1.20 nm, which is similar to the pore size of pristine COF membrane. This suggests that the chain-like MOFs may not fully occupy the COF pore channels in a complete top-down filling manner (Supplementary Figure 7f).

We also conducted simulations on the pore diameters of CMOF and COF. It was found that the pore diameter of COF membrane is approximately 1.17 nm. After introducing MOFs, the pore diameter decreases to 0.67 nm, which is basically consistent with the experimental characterization results. This discussion has been added to the revised manuscript.

Figure 7. Simulated pore size distributions of pristine COFs (c) and 1D MOF-in-2D COF hetero-structure (derived from **Figure 7e**) (d). The simulated 1D MOF-in-2D COF hetero-structure (Some atoms are omitted for clarity.) (e).

Original: The TAPA-TFP-0.25-NH₂-CuBDC CMOF composite membrane features a main pore diameter of 0.68 nm, which is smaller than that of COFs (1.22 nm) or MOFs (1.42 nm),....

Revised: The TAPA-TFP-0.25-NH₂-CuBDC CMOF composite membrane features a main pore diameter of 0.68 nm (basically consistent with simulation results), which is smaller than that of COFs (1.22 nm) or MOFs (1.42 nm),...

Furthermore, in the manuscript, we did not mention that MOF and COF are interlocked structures. Instead, MOF or COF undergo interlocking growth process with the help of some reaction functional groups (“Concretely, by interlocking and in situ immobilized growth, the pristine COF membrane is capable of capturing MOF ligands and metal ions in sequence to form 1D MOF-in-2D COF hetero-structured composite membranes.”). The interlocking growth we proposed in the manuscript is an illustrative description of the membrane growth process and mechanism, which is fundamentally different from the interlocked COF reported in the literature.

6. Please provide HRTEM images of pure MOFs and COFs as references.

Answer: Thank you very much for your valuable suggestions. HRTEM images of pure MOFs (Supplementary Figure 6a) and COFs (Supplementary Figure 6b) were provided in the supporting information.

Figure 6. TEM images of pristine MOFs (0.25-NH₂-CuBDC MOFs) (a), the pristine COF membrane (0.5 mM TAPA + 0.5 mM TFP) (b) and the TAPA-TFP-0.25-NH₂-CuBDC CMOF membrane (0.5 mM TAPA + 0.5 mM TFP) (b) and the TAPA-TFP-0.25-NH₂-CuBDC CMOF

7. Since it is possible that MOF growth might disrupt original COF stacking due to missing peaks in XRD data, how did the authors ensure an ideal 1D MOFs within 2D COFs structure?

Answer: Thank you very much for your valuable suggestions. It should be noted that, during the process of preparing the CMOF membrane, the ordered and definite structure of the COF membrane has been established and ensured before (as shown in the following figure). On this basis, the ligands and metal for forming MOF are successively introduced to construct the CMOF structure. The XRD characteristic peaks from COFs that should have been detected in the CMOF structure were not detected. The reasons maybe in two aspects as follows: (1) When the XRD peaks of MOF and COF coexist, the peak from MOF will significantly compress that of COF, making it impossible to detect. (2) The structure of MOF within the ordered COF pores masks the lattice signal of COF.

The ordered and definite structure of the COF membrane had already been ensured before forming the CMOF membrane. If we want to confirm the actual structure of CMOF, the most important step is to examine the state of MOF within the COF channels (the detected XRD signal from MOFs matters).

As shown in the figure below, the XRD signal of MOF is significantly stronger than that of COF. It should be noted that even after reducing the MOF signal intensity by fivefold when processing the data for plotting, it remains stronger than the XRD signal of COF. This means that when MOF and COF coexist, the intense XRD signal from MOF will overwhelmingly suppress the COF signals, making the XRD peaks of COF extremely difficult to observe. Furthermore, as shown in the figure below, the (110) crystallographic plane of the COF can be observed, but the signal is weaker, which may be attributed to the introduction of the MOF reducing the crystallinity and masking the (100) crystal plane (making it difficult to be detected) of COF. Meanwhile, the right-shift of the (110) crystallographic plane confirms the effective growth of MOF within the COF channels and the relatively strong interaction between MOF and COF. This discussion has been added to the revised manuscript.

Original: It is worth mentioning that the sharp XRD peaks of the pristine COF membrane are not distinctly observed in the CMOF composite membrane, which may result from the introduction of MOFs into the pore structure or interlayer of COFs based on coordination interactions disrupting the long-range periodicity of the porous structure of COFs^{17,23}.

Revised: It is worth mentioning that the sharp XRD peaks of the pristine COF membrane that should have been detected (The ordered and definite structure of the COF membrane had already been established before forming the CMOF membrane as displayed in **Figure 2a.**) are not distinctly observed in the CMOF composite membrane, which may result from: 1) the introduction of MOFs into the pore structure or interlayer of COFs based on coordination interactions masking the XRD signals from COFs, and 2) the intense XRD signals from MOFs overwhelmingly suppressing the COF signals, making the XRD peaks from COFs extremely difficult to observe when MOFs and COFs coexist^{17,23}. Still, the (110) crystallographic plane with weak signals of pristine COF membrane can be observed in CMOF composite membrane (**Figure 2f**), and its right-shift compared with pristine COF membrane confirms the effective growth of MOFs within the COF channels and the relatively strong interaction between MOFs and COFs.

Furthermore, to address this issue, we conducted TEM characterization and identified distinct crystalline domains and lattice fringes corresponding to COF and MOF respectively, thereby confirming their coexistence (as shown in the figure below).

In addition, to clarify the actual structure of 1D MOF-in-2D COF as closely as possible, we conducted structural simulations to visualize the 1D MOF-in-2D COF hetero-structure based on the characterization results. As shown in the following figure, the distance between Cu center from MOFs and its neighboring N atom from COFs ranges from 0.28 to 0.73 nm, which confirms the short-range coordination interaction between MOFs and COFs. This discussion has been added to the revised manuscript.

Figure 1. (f) The simulated 1D MOF-in-2D COF hetero-structure (Some atoms are omitted for clarity.). The distance between Cu center from MOFs and its neighboring N atom from COFs ranges from 0.28 to 0.73 nm, which confirms the short-range coordination interaction between MOFs and COFs.

Added: Simulation results indicate the short-range coordination interaction between MOFs and COFs (**Supplementary Figure 1f**).

In addition to the 0.68 nm channel size from BET analysis, another peak is observed, for which we have provided a detailed explanation regarding its emergence in the revised manuscript (as shown in the figure below).

Added: It is noteworthy that the CMOF composite membrane exhibits another weaker peak at 1.20 nm, which is similar to the pore size of pristine COF membrane. This suggests that the chain-like MOFs may not fully occupy the COF pore channels in a complete top-down filling manner (**Supplementary Figure 7f**).

Figure 7. Schematic diagram of CMOF pore channels (f).

We also conducted simulations on the pore diameters of CMOF and COF. It was found that the pore diameter of COF membrane is approximately 1.17 nm. After introducing MOFs, the pore diameter decreases to 0.67 nm, which is basically consistent with the experimental characterization results. This discussion has been added to the revised manuscript.

Figure 7. Simulated pore size distributions of pristine COFs (c) and 1D MOF-in-2D COF hetero-structure (derived from **Figure 7e**) (d). The simulated 1D MOF-in-2D COF hetero-structure (Some atoms are omitted for clarity.) (e).

Original: The TAPA-TFP-0.25-NH₂-CuBDC CMOF composite membrane features a main pore diameter of 0.68 nm, which is smaller than that of COFs (1.22 nm) or MOFs (1.42 nm),.....

Revised: The TAPA-TFP-0.25-NH₂-CuBDC CMOF composite membrane features a main pore diameter of 0.68 nm (basically consistent with simulation results), which is smaller than that of COFs (1.22 nm) or MOFs (1.42 nm),.....

In the revised manuscript, the relevant statements have been corrected.

Original: It is worth mentioning that the sharp XRD peaks of the pristine COF membrane are not distinctly observed in the CMOF composite membrane, which may result from the introduction of MOFs into the pore structure or interlayer of COFs based on coordination interactions disrupting the long-range periodicity of the porous structure of COFs^{17,23}.

Revised: It is worth mentioning that the sharp XRD peaks of the pristine COF membrane that should have been detected (The ordered and definite structure of the COF membrane had already been established before forming the CMOF membrane as displayed in **Figure 2a**.) are not distinctly observed in the CMOF composite membrane, which may result from: 1) the introduction of MOFs into the pore structure or interlayer of COFs based on coordination interactions masking the XRD signals from COFs, and 2) the intense XRD signals from MOFs overwhelmingly suppressing the COF signals, making the XRD peaks from COFs extremely difficult to observe when MOFs and COFs coexist^{17,23}.

Based on the above experimental characterizations and simulation results, we believe that the actual structure of CMOF is relatively very clear. On the contrary, the MCOF membrane has almost no ion separation capability. We believe that in this case, the COF structure may have been damaged and the MCOF structure is relatively more complex. Benefiting from the pre-formation of the COF skeleton, the structure of CMOF is very clear.

8. Please explain why there is a tenfold decrease in K^+/Na^+ selectivity within binary systems

Answer: Thank you very much for your valuable suggestions. In the manuscript, we did not mention a tenfold decrease in K^+/Na^+ selectivity within binary systems. We find “The K^+ permeation rate of TAPA-TFP-0.25- NH_2 -CuBDC CMOF composite membrane (COFs: 1 mM TAPA + 1 mM TFP) presents a proportional rise to the concentration gradient, but K^+/Na^+ selectivity may show a nearly 10-fold decrease if the concentration gradient is low (**Figure 4h**).”. This 10-fold decrease in selectivity remains an ideal selectivity. The reason for the decrease is that during the test, the ion concentration measured in the feed side decreased from 0.1 M to 0.02 M, and the lower ion concentration resulting in a lower osmotic pressure and driving force, which weakened the transport barrier difference of ions.

The K^+/Na^+ selectivity within binary systems has indeed decreased to some extent, dropping from 82 to 51. However, it is believed to be normal (<https://doi.org/10.1038/s44221-024-00379-3>, Nature Water, 2025, 191-200: Li^+/Mg^{2+} selectivity decreases from 203 to 81) due to the competition for occupying effective mass transport channels between cations. MD simulations were also conducted to confirm this phenomenon. The information has been added to the revised manuscript.

9. While pore entrance energy barriers can be supported by membrane pore size data, please also provide evidence for intrapore energy barriers. Please specify transport channels within this MOFs-COFs composite so authors may identify functional groups responsible for affecting transport since carboxyl is not solely present in composites.

Answer: Thank you very much for your professional suggestions. Following your suggestion, molecular dynamics simulations are conducted to explore the ion transport paths, energy barriers, and interactions with the 1D MOF-in-2D COF channels in the revised manuscript.

Figure 5 | K⁺-selective transport mechanism. **a**, Schematic diagram of ion transport process. **b-c**, The interaction energy obtained from density functional theory (DFT) calculations. **d**, Simulation snapshots visually depict the transmembrane behavior of Na⁺ (yellow) and K⁺ ions (pink) (C, O, N, H, and Cu are depicted in grey, red, blue, white and orange respectively.). **e**, PMF profiles for K⁺, Na⁺, Li⁺ and Mg²⁺ ions across the CMOF membrane. **f**, Transmembrane ion permeation kinetics across the CMOF membrane in single-ion and binary-ion systems.

Added: The CMOF membrane exhibits two different-sized pores (**Figure 2e**). The 0.68-nanometer pores are dedicated to effective ion screening, while the 1.2-nanometer pores facilitate rapid ion migration (**Figure 5a**).

Added: Molecular dynamics (MD) simulations also confirm the selective transport of K⁺ ions over other metal ions (especially Na⁺ ions) in the CMOF channels, which is supported by the snapshots showing the transmembrane migration behavior of Na⁺ (yellow) and K⁺ ions (pink) through the CMOF membranes (**Figure 5d**), and the ion transmembrane energy barrier (**Figure 5e**). As displayed in **Figure 5e**, Potentials of mean force (PMF) profiles exhibit a gradual increase in energy as all ions approach the membrane pores, and K⁺ ions face the

lowest energy barriers during transmembrane migration, thereby facilitating K^+ ion transport and rapid transmembrane (The transmembrane energy barrier for Mg^{2+} is the highest ($34.51 \text{ kcal mol}^{-1}$), which is much greater than that of Li^+ ($18.73 \text{ kcal mol}^{-1}$), Na^+ ($15.56 \text{ kcal mol}^{-1}$), and K^+ ($8.12 \text{ kcal mol}^{-1}$)). Furthermore, during the same period, the number of K^+ ions passing through the CMOF membrane is significantly higher than that of Na^+ ions, indicating superior K^+/Na^+ ideal selectivity (**Figure 5f**), which is consistent with experimental results (confirming the efficacy of CMOF membranes in ion separation and lower actual selectivity). These findings suggest that the excellent permselectivity of the CMOF membrane is attributed to the synergistic effect of pore-entrance size-sieving and in-pore transport dynamics.

Added: Molecular dynamics (MD) simulations. MD simulations were conducted using the GROMACS 2021.7 program, with subsequent analysis carried out using VMD 1.9.3. LJ parameters for atoms within the CMOFs were obtained from the UFF4MOF2, and the OPC3 model was used to describe water. The REPEAT charges of CMOFs were calculated using CP2K under the PBE-D3/TZV2P-MOLOPT-GTH method. The CMOF membrane was positioned in the middle-right region of a simulation box with dimensions of $5.8 \text{ nm} \times 3.4 \text{ nm} \times 26 \text{ nm}$. A specific number of water molecules and ions were added to the system. For the simulation of a binary salt system comprising KCl and NaCl at a concentration of 1 M each, we included 6890 water molecules, 104 K^+ ions, and 104 Na^+ ions. Cl^- ions were added to neutralize the overall charge of the system, with an additional 5673 water molecules introduced on the permeate side. The simulation protocol involved energy minimization using the steepest descent method, followed by 200 ps of pre-equilibration at 300 K. Subsequently, a 150 ns equilibrium molecular dynamics simulation run was performed in the canonical ensemble (NVT), with the system temperature controlled at 300 K using the Nose-Hoover method. Cut-off distances were set to 1.2 nm, and the PME method was utilized to compute long-range electrostatic interactions. Periodic boundary conditions (PBC) were applied in two dimensions to minimize edge effects. Potentials of mean force (PMF) profiles for ions traversing the membrane were determined using umbrella sampling. Thirty-two windows were evenly distributed along the z-axis, covering a range from 1.0 to 2.6 nm for the ions, with each window undergoing an 8.0 ns simulation run.

10. Control samples like pure COFs membranes should be investigated to demonstrate the role of MOFs. Answer: Thank you very much for your professional suggestions. The information about pure COFs membranes were provided in the supporting information, such as, pore size (Supplementary Figure 7b), cation diffusion behavior (Supplementary Figure 35).....

Figure 7. Pore size distributions and Brunauer-Emmett-Teller (BET) surface areas of pristine MOFs (a) and COF membranes (b). The pristine MOFs and COF membrane respectively exhibit

Figure 35. Ion conductivity in the permeate side as a function of time for the pristine COF membranes (a, COFs: 0.5 mM TAPA + 0.5 mM TFP. b, COFs: 1 mM TAPA + 1 mM TFP) and PES substrate (c). It should be noted that the pristine COF membrane and PES substrate have no ion selectivity.

Response to Reviewer #2:

Comments:

The manuscript reports a biomimetic KcsA channel based on a 1D MOF-in-2D COF hetero-structured composite membranes for highly selective ion separation. The composite membranes are formed via interlocking and in situ immobilized growth, employing a bottom-up strategy. The composite membranes present a molecular-level interlinked hybridization of covalent and metal organic hetero-frameworks. The resulting CMOF composite membrane with biomimetic KcsA channels allows precise K⁺ recognition transport. Pore-entrance sieving effect and channel wall-ion interactions enable ultra-selective and fast transport of K⁺ with unprecedented K⁺/Na⁺ selectivity of 10² and K⁺/Mg²⁺ selectivity of 10³. I think this manuscript is suitable for publication in Nature Communication, provided that the authors consider the following comments:

1. The manuscript mentions that the 2D TAPA-TFP COF membranes have superior tenacity and flexibility, yet no mechanical property tests are reported. Relevant mechanical testing, such as tensile tests, is recommended to support these claims.

Answer: Thank you very much for your professional suggestions. Following your suggestion, in the revised manuscript, relevant mechanical testing (tensile tests) are included. Furthermore, the video demonstrating the excellent toughness of the CMOF membrane has also been included in the revised manuscript.

Figure 2. (b) Force-elongation curves tested using a microcomputer-controlled electronic tensile testing machine. CMOF membranes demonstrate superior mechanical properties compared to pristine COF membranes. Furthermore, as the MOF ligand concentration increases, the mechanical properties of CMOF membranes tends to decline.

Added: The tensile properties were measured by an electronic tensile testing machine (DEBEN5000DL) with a tensile rate of 0.1 mm/min controlled by a microcomputer (In an aqueous solution, a circular membrane formed in the petri dish is folded twice (first in half, then again in half), resulting in a composite membrane with four layers tightly stacked together. Subsequently, the composite membrane is carefully lifted out from the aqueous solution using a substrate and allowed to air-dry naturally. The dried membrane is then trimmed into 40 mm × 10 mm strips for mechanical tensile testing.).

Original: The TAPA-TFP-x-NH₂-CuBDC CMOF membrane shows a Janus morphology and its bottom surface is uniformly distributed with broken thin COF vesicles that nearly occupy the surface, in contrast to the pristine COF membrane (almost identical top and bottom surface morphologies) (**Figure 2c**, **Supplementary Figure 2**).

Revised: The TAPA-TFP-x-NH₂-CuBDC CMOF membrane with superior mechanical properties shows a Janus morphology and its bottom surface is uniformly distributed with broken thin COF vesicles that nearly occupy the surface, in contrast to the pristine COF membrane (almost identical top and bottom surface morphologies) (Figure 2c, Supplementary Figure 2, Supplementary Video 1).

2. The description of the interfacial polymerization process is too brief. For example, during CMOF membrane preparation, key details such as the injection rate of NH₂-BDC and Cu(NO₃)₂, the injection angle, and the control of interfacial stability are not fully explained. These factors can significantly impact membrane homogeneity and performance. The authors should provide more detailed experimental procedures.

Answer: Thank you very much for your professional suggestions. Following your suggestion, more detailed experimental procedures during interfacial polymerization process are included in the revised manuscript.

Original: After the COF membrane was successfully prepared at the interface, NH₂-BDC dissolved in DCM (NH₂-BDC was first dissolved with DMF and then dissolved in DCM) was slowly injected into the underlying oil phase by an injection syringe. One day later, Cu(NO₃)₂ aqueous solution was slowly injected into the upper aqueous phase.

Revised: After the COF membrane was successfully prepared at the interface, NH₂-BDC dissolved in DCM (NH₂-BDC was first dissolved with DMF and then dissolved in DCM) was slowly injected into the underlying oil phase by an injection syringe (The syringe needle tip is perpendicular to the liquid surface with the injection rate of 1 mL/min.). One day later, Cu(NO₃)₂ aqueous solution was slowly injected into the upper aqueous phase (The syringe needle tip is perpendicular to the liquid surface with the injection rate of 0.5 mL/min.). It should be noted that, if the injection rate of NH₂-BDC and Cu(NO₃)₂ is too high, it may generate excessive bubbles and damage the integrity of the COF membrane at the interface, which will significantly affect the morphology and separation performance of the CMOF membrane.

Added: In the whole process, the operation should be gentler than during the preparation of CMOF membranes because the MOFs at the interface is merely loosely assembled through van der Waals forces rather than forming a resilient membrane, making the MOFs more prone to random migration, which will ultimately compromise the quality of MCOF membranes. The syringe needle tip is perpendicular to the liquid surface with the injection rate of 0.3 mL/min when introducing TAPA and TFP.

3. Further studies, including multiple repeat experiments and continuous operation over days or weeks, are recommended to demonstrate the stability of the membrane performance. Additionally, the membrane's stain resistance tests should be conducted to evaluate.

Answer: Thank you very much for your valuable suggestions. Following your suggestion, to demonstrate the stability and stain resistance of the membrane, multiple repeat experiments and continuous operation over days are conducted and the results are added to the revised manuscript.

Original: After 20 hours of testing, the membrane still maintains excellent integrity, toughness, and K⁺-selective transport property (Supplementary Figure 36).

Revised: After a long period of testing, the membrane still maintains excellent integrity, toughness, and K⁺-selective transport property (Supplementary Figure 36).

Figure 36. (a) TAPA-TFP-0.25-NH₂-CuBDC CMOF composite membranes (COFs: 1 mM TAPA + 1 mM TFP) after 20 hours of testing. (b-c) The K⁺ and Na⁺ diffusion behavior in TAPA-TFP-0.25-NH₂-CuBDC CMOF composite membranes (COFs: 1 mM TAPA + 1 mM TFP) for long-term stability test. For Figure 36c, the COF membranes were tested for 7 hours every day, with samples being taken and tested at the 1st, 3rd, and 7th hours respectively. Then the COF membranes were washed, and immersed in pure water and left to stand for the test on the next day.

4. The manuscript explains the ion selective transport mechanism through experimental data and basic theoretical analysis. It is recommended that the authors incorporate molecular dynamics simulations to explore the ion transport paths, energy barriers, and interactions with the channel walls in the 1D MOF-in-2D COF channels.

Answer: Thank you very much for your professional suggestions. Following your suggestion, molecular dynamics simulations are conducted to explore the ion transport paths, energy barriers, and interactions with the 1D MOF-in-2D COF channels, and the results are added to the revised manuscript.

Figure 5 | K^+ -selective transport mechanism. **a**, Schematic diagram of ion transport process. **b-c**, The interaction energy obtained from density functional theory (DFT) calculations. **d**, Simulation snapshots visually depict the transmembrane behavior of Na^+ (yellow) and K^+ ions (pink) (C, O, N, H, and Cu are depicted in grey, red, blue, white and orange respectively.). **e**, PMF profiles for K^+ , Na^+ , Li^+ and Mg^{2+} ions across the CMOF membrane. **f**, Transmembrane ion permeation kinetics across the CMOF membrane in single-ion and binary-ion systems.

Added: The CMOF membrane exhibits two different-sized pores (**Figure 2e**). The 0.68-nanometer pores are dedicated to effective ion screening, while the 1.2-nanometer pores facilitate rapid ion migration (**Figure 5a**).

Added: Molecular dynamics (MD) simulations also confirm the selective transport of K^+ ions over other metal ions (especially Na^+ ions) in the CMOF channels, which is supported by the snapshots showing the transmembrane migration behavior of Na^+ (yellow) and K^+ ions (pink) through the CMOF membranes (**Figure 5d**), and the ion transmembrane energy barrier (**Figure 5e**). As displayed in **Figure 5e**, Potentials of mean force (PMF) profiles exhibit a gradual increase in energy as all ions approach the membrane pores, and K^+ ions face the

lowest energy barriers during transmembrane migration, thereby facilitating K^+ ion transport and rapid transmembrane (The transmembrane energy barrier for Mg^{2+} is the highest ($34.51 \text{ kcal mol}^{-1}$), which is much greater than that of Li^+ ($18.73 \text{ kcal mol}^{-1}$), Na^+ ($15.56 \text{ kcal mol}^{-1}$), and K^+ ($8.12 \text{ kcal mol}^{-1}$)). Furthermore, during the same period, the number of K^+ ions passing through the CMOF membrane is significantly higher than that of Na^+ ions, indicating superior K^+/Na^+ ideal selectivity (**Figure 5f**), which is consistent with experimental results (confirming the efficacy of CMOF membranes in ion separation and lower actual selectivity). These findings suggest that the excellent permselectivity of the CMOF membrane is attributed to the synergistic effect of pore-entrance size-sieving and in-pore transport dynamics.

Added: **Molecular dynamics (MD) simulations.** MD simulations were conducted using the GROMACS 2021.7 program, with subsequent analysis carried out using VMD 1.9.3. LJ parameters for atoms within the CMOFs were obtained from the UFF4MOF2, and the OPC3 model was used to describe water. The REPEAT charges of CMOFs were calculated using CP2K under the PBE-D3/TZV2P-MOLOPT-GTH method. The CMOF membrane was positioned in the middle-right region of a simulation box with dimensions of $5.8 \text{ nm} \times 3.4 \text{ nm} \times 26 \text{ nm}$. A specific number of water molecules and ions were added to the system. For the simulation of a binary salt system comprising KCl and NaCl at a concentration of 1 M each, we included 6890 water molecules, 104 K^+ ions, and 104 Na^+ ions. Cl^- ions were added to neutralize the overall charge of the system, with an additional 5673 water molecules introduced on the permeate side. The simulation protocol involved energy minimization using the steepest descent method, followed by 200 ps of pre-equilibration at 300 K. Subsequently, a 150 ns equilibrium molecular dynamics simulation run was performed in the canonical ensemble (NVT), with the system temperature controlled at 300 K using the Nose-Hoover method. Cut-off distances were set to 1.2 nm, and the PME method was utilized to compute long-range electrostatic interactions. Periodic boundary conditions (PBC) were applied in two dimensions to minimize edge effects. Potentials of mean force (PMF) profiles for ions traversing the membrane were determined using umbrella sampling. Thirty-two windows were evenly distributed along the z-axis, covering a range from 1.0 to 2.6 nm for the ions, with each window undergoing an 8.0 ns simulation run.

5. More relevant membranes related works should be cited, such as Yang et al., Science 2019, 364, 1057-1062.; Wang et al., ACS Nano 2024, 18, 34698-34707.; Hong et al., Nature Commun. 2024, 15:3160.

Answer: Thank you very much for your valuable suggestions. Following your suggestion, the relevant membranes related works are cited in the revised manuscript, including Yang et al., Science 2019, 364, 1057-1062.; Wang et al., ACS Nano 2024, 18, 34698-34707.; Hong et al., Nature Commun. 2024, 15:3160.